

# CSA-DE-LR: enhancing cardiovascular disease diagnosis with a novel hybrid machine learning approach

Beyhan Adanur Dedeturk[1], Bilge Kagan Dedeturk[2] and Burcu Bakir-Gungor[1]

[1] Department of Computer Engineering, Abdullah Gul University, Kayseri, Turkey
[2] Department of Software Engineering, Erciyes University, Kayseri, Turkey

## ABSTRACT

Cardiovascular diseases (CVD) are a leading cause of mortality globally, necessitating the development of efficient diagnostic tools. Machine learning (ML) and metaheuristic algorithms have become prevalent in addressing these challenges, providing promising solutions in medical diagnostics. However, traditional ML approaches often need to be improved in feature selection and optimization, leading to suboptimal performance in complex diagnostic tasks. To overcome these limitations, this study introduces a new hybrid method called CSA-DE-LR, which combines the clonal selection algorithm (CSA) and differential evolution (DE) with logistic regression. This integration is designed to optimize logistic regression weights efficiently for the accurate classification of CVD. The methodology employs three optimization strategies based on the F1 score, the Matthews correlation coefficient (MCC), and the mean absolute error (MAE). Extensive evaluations on benchmark datasets, namely Cleveland and Statlog, reveal that CSA-DE-LR outperforms state-of-the-art ML methods. In addition, generalization is evaluated using the Breast Cancer Wisconsin Original (WBCO) and Breast Cancer Wisconsin Diagnostic (WBCD) datasets. Significantly, the proposed model demonstrates superior efficacy compared to previous research studies in this domain. This study's findings highlight the potential of hybrid machine learning approaches for improving diagnostic accuracy, offering a significant advancement in the fields of medical data analysis and CVD diagnosis.

## INTRODUCTION

Individuals follow a daily routine and maintain a busy schedule, leading to stress and concern. Moreover, there has been a significant rise in the prevalence of cigarette addiction and obesity, contributing to the surge in diseases such as cancer, cardiac issues, and various other health conditions (*Pouriyeh et al., 2017*). The most formidable aspect of these illnesses lies in their predictability. Anticipating the onset of these diseases poses a considerable challenge. A staggering reality is revealed by estimations from the World Health Organization (WHO): approximately 17.9 million lives are lost annually to cardiovascular diseases, underscoring the alarming fact that nearly 32% of global

Corresponding author
Beyhan Adanur Dedeturk,
beyhan.adanur@agu.edu.tr

fatalities are attributed to heart-related issues (*World Health Organization, 2021*). Within the spectrum of cardiovascular diseases, one notably challenging and prevalent condition is coronary artery disease (CAD).

CAD manifests when the coronary arteries, which supply the heart with its essential blood requirements, become obstructed. The accumulation of cholesterol and other substances within these arteries leads to plaque formation, gradually filling the vessels and impeding blood flow. In its early stages, this arterial narrowing may manifest as chest pain. However, diagnosing CAD poses a significant challenge, often resulting in severe symptoms such as heart attacks or heart failure becoming the primary indicators for patients.

The initial step in ascertaining the presence of CAD involves assessing whether a patient falls within the high-risk category. Once identified as high-risk, a battery of tests, including but not limited to blood tests, chest X-rays, coronary angiograms, electrocardiograms, and echocardiograms, is administered (*CDC, 2021*). These diagnostic procedures, while crucial, are not only intricate but also incur substantial costs, contributing to the complexity and expense associated with the identification and management of CAD. Thus, the imperative for ongoing research and innovative approaches in the medical field persists to enhance both the accuracy and accessibility of cardiovascular disease diagnoses.

The application of machine learning (ML) is widely endorsed for cardiovascular disease prediction, given its proficiency in extracting exceptionally efficient and precise data from extensive datasets, streamlining the prediction process (*Alkayyali, Idris & Abu-Naser, 2023*; *Azmi et al., 2022*). As the fundamental tenet of ML, it excels in managing substantial data volumes, demonstrating swift processing capabilities, and furnishing predictions at the initial stages of development. ML applications are pivotal in alleviating hospital errors and propelling advancements in health policy, disease prevention, early detection, and reducing avoidable hospital fatalities. Several studies have undertaken similar objectives, specifically delineating ML approaches adept at diagnosing CAD (*Naser et al., 2024*).

Initially, ML algorithms like logistic regression (LR), XGBoost, support vector machine (SVM), and Naive Bayes (NB) were used for CVD prediction (*Kolukısa et al., 2019*; *Kolukisa et al., 2020*; *Kolukisa & Bakir-Gungor, 2023*; *Dhanka, Bhardwaj & Maini, 2023*), but they struggle with handling complex and multidimensional data, leading to lower success rates (*Ramudu et al., 2023*). The local minima problem, a major obstacle to traditional approaches, affects convergence to optimum solutions. To minimize the objective function, model parameters are updated iteratively using gradient-based optimization techniques like gradient descent. However, if initialization or parameter updates push algorithms into less-than-ideal solutions, they may become stuck in local minima (*Ghassemi et al., 2020*). In the literature, metaheuristics have been employed by researchers to select features, to optimize parameters, and to train the standard ML algorithms for improving their classification accuracy by avoiding local minima.

Most studies in the literature have used metaheuristic approaches for feature selection (to reduce dimensionality and speed up computation time) and hyperparameter optimization (to find the nearly optimal configurations for ML models) problems without focusing on training ML algorithms (*Nalluri et al., 2017*; *Murugesan et al., 2021*; *Muliawan, Rizal & Hadiyoso, 2023*; *Torthi et al., 2024*; *Sampathkumar & Periyasamy, 2024*; *Dhanka & Maini,*

*2024*). These works are distinct from problems of the proposed method since in this study, metaheuristics are used to train the LR while maintaining their good qualities.

In a few research works, metaheuristics have been employed by researchers to train the standard ML algorithms for improving their classification accuracy by avoiding local minima (*Leema, Nehemiah & Kannan, 2016*; *Arabasadi et al., 2017*; *Poornima & Gladis, 2018*; *Shahid & Singh, 2020*; *Al Bataineh & Manacek, 2022*). These hybrid algorithms, which combine metaheuristics and machine learning algorithms, have shown superior performance in diagnosing cardiovascular disease (CVD) compared to other techniques. However, these methods require significant time and effort to improve detection rates and classification performance. To minimize these disadvantages of current hybrid methods, the ML method to be used must first be computationally efficient for large datasets and have simplicity-interpretability in disease diagnosis problems. Afterwards, it is necessary to choose the metaheuristic algorithm that will best overcome the disadvantages of the ML method to be used and will be successful for the train and suitable for the relevant problem (*Naser et al., 2024*).

An important artificial immune system (AIS) method called the clonal selection method (CSA) produces antibodies with increased affinity over time, hence enhancing the immune system's response to antigens. CSA is widely used in optimization problems due to its ability to employ receptor editing and hyper-mutation processes to explore the solution space for both local and global solutions (*Rahman et al., 2023*). Additionally, research have shown that, in a variety of scenarios, CSA-based strategies perform better than other bio-inspired and optimization methods (*Haktanirlar Ulutas & Kulturel-Konak, 2011*; *Duru et al., 2022*; *Rahman et al., 2023*). However, there is a chance that conventional CSA methods aren't providing enough search power and might use some refinement (*Gong, Jiao & Zhang, 2010*; *Zhang et al., 2008*; *Xu et al., 2017*). Thus, by utilizing techniques that are known to have exceptional search performance, such as the differential evolution (DE) algorithm, a hybrid optimization approach may be built to enhance the local search performance of the CSA method (*Mostafa et al., 2024*; *Azevedo, Rocha & Pereira, 2024*; *Song et al., 2024*).

In this study, a novel hybrid classifier called as CSA-DE-LR is proposed in an attempt to improve the poor detection rate in CVD prediction and address the inadequacies of previous studies (*Naser et al., 2024*). Because LR is easily interpretable in illness diagnostic issues and computationally economical for big data sets, it is used as a classification model in this work for the diagnosis of CVD. Then, CSA-DE optimization method employees for model training instead of the gradient descent algorithm to improve LR's classification accuracy by avoiding local minima (*Dedeturk & Akay, 2020*; *Dedeturk, Akay & Karaboga, 2021*). Based on the F1 score, the Matthews correlation coefficient (MCC), and the mean absolute error (MAE), the technique uses three optimization strategies. CSA-DE-LR outperforms state-of-the-art ML methods, according to extensive evaluations on benchmark datasets which include Cleveland and Statlog. In addition, generalization tests are evaluated using the Breast Cancer Wisconsin Original (WBCO) and Breast Cancer Wisconsin Diagnostic (WBCD) datasets. Significantly, the proposed model demonstrates superior efficacy compared to previous research studies in this domain. The results of this study demonstrate how hybrid ML techniques may improve diagnostic precision and

represent a substantial breakthrough in the domains of medical data analysis and CVD diagnosis. The main contributions of this study are as follows:

- **Introducing a breakthrough in classification methodology:** This article presents CSA-DE-LR, a pioneering classification methodology that merges a clonal selection algorithm (CSA) and differential evolution (DE) with LR. This innovative hybrid approach is tailored to enhance LR weights for efficient classification, particularly in the context of CVD. Most studies in the literature have used meta-heuristic approaches for the problems of feature selection and hyperparameter optimization without focusing on training ML algorithms. Unlike these studies, metaheuristics are used to train the ML algorithms for the proposed method. It also provides detailed information about the rationale behind combining these three specific methodologies.
- **Implementation of three optimization techniques:** The proposed CSA-DE-LR method offers three distinct optimization strategies based on the F1 score, MCC, and MAE. These metrics guide the training process and are critical in fine-tuning the model weights for optimal classification performance.
- **Comprehensive evaluation of the method:** The study extensively evaluates the CSA-DE-LR method using two well-known datasets: the Cleveland and Statlog datasets. The performance is assessed using a range of metrics, including accuracy, F1 score, MCC, ROC-AUC, false negative rate, and false positive rate, and the results are compared with various popular machine learning techniques. Care is taken to be fully transparent and fair when comparing the proposed method with other studies and displaying the results. In addition, generalization tests are evaluated using the WBCO and WBCD datasets. The ethical implications of using ML models in healthcare are also evaluated.
- **Insights into feature selection and model optimization:** Unlike current studies, the article provides valuable insights into the impact of feature selection and model optimization through detailed analysis. It explores how excluding certain features can lead to improved predictive consistency and generalizability, highlighting the importance of dataset-specific tuning and careful consideration of feature selection with a different approach.
- **Advancement in classification performance:** The article shows that CSA-DE-LR outperforms previous methods in terms of accuracy and precision on both Cleveland and Statlog datasets. This demonstrates the method's effectiveness and potential in improving diagnostic decision-making processes in the medical field.

This article introduces CSA-DE-LR, a novel method for classifying cardiovascular diseases. The 'Methods' section explains how the Clonal Selection Algorithm, Differential Evolution, and Logistic Regression are integrated into CSA-DE-LR. The 'Experiments' section outlines the datasets utilized, the evaluation metrics applied, and the hyperparameter optimization process. The 'Discussion' section explores the implications of the proposed model findings, focusing on feature selection and the effectiveness of CSA-DE-LR in diagnosing cardiovascular diseases. Finally, the 'Conclusions' section summarizes this research, emphasizing the significance and potential applications of the CSA-DE-LR method in the medical field.

# RELATED WORK

Multiple investigations are currently underway in the field of cardivascular disease diagnosis, each serving distinct objectives (*Naser et al., 2024*). These investigations are driven by the need to enhance diagnostic accuracy and efficiency. These objectives include identifying optimal features crucial for accurate diagnosis, developing innovative classification models tailored to the complexities of cardiac conditions, and creating highly efficient classification methods capable of streamlining the diagnostic process. Various methods, encompassing ML techniques and metaheuristic-based approaches, have been proposed to achieve comprehensive and effective diagnostic outcomes (*Cai et al., 2024*; *Rani et al., 2024*).

## Machine learning techniques

In the initial research works, ML algorithms such as LR, XGBoost, SVM, NB, *etc.* and similar methods were used for CVD prediction (*Kolukısa et al., 2019*; *Kolukisa et al., 2020*; *Kolukisa & Bakir-Gungor, 2023*; *Dhanka, Bhardwaj & Maini, 2023*).

*Kolukısa et al. (2019)* expanded the range of feature selection methodologies to enhance performance. Additionally, Fisher linear discriminant analysis was applied to reduce computational time by decreasing the number of features in diagnosing coronary artery disease, resulting in well-performing models for each dataset. Utilizing the MLP classifier, they achieved an accuracy of 82.5% and an F-measure of 83.80% on the Cleveland dataset. In *Kolukisa et al. (2020)*, a novel self-optimized and adaptive ensemble ML algorithm was introduced. This algorithm autonomously identifies the most appropriate machine learning models, ensuring high accuracy across diverse coronary artery disease datasets. On the Cleveland dataset, an accuracy of 83.43% was achieved. Furthermore, Kolukisa and Bakir-Gungor's work (*2023*), which incorporated the Z-Alizadehsani, Cleveland, and Statlog datasets, proposed an exhaustive ensemble feature selection (FS) method and a probabilistic ensemble FS approach. The evaluation encompassed six distinct classification algorithms and four variants of voting algorithms. The obtained accuracy values were 85.47% for the Cleveland dataset and 85.55% for the Statlog dataset.

*Dhanka, Bhardwaj & Maini (2023)* present a thorough examination of LR and XGBoost in the Statlog heart disease dataset as a benchmark. Model parameters are optimized through Random SearchCV hyperparameter tuning. The study encompasses an analysis of both non-optimized and optimized models. The results derived from 10-fold cross-validation indicate that LR and XGBoost accuracies are 85.2% and 81.5%, respectively.

Although standard ML techniques have shown notable success in CVD prediction, they often struggle with handling complex and multidimensional data, which can lead to lower success rates (*Ramudu et al., 2023*). Furthermore, the local minima problem presents a major obstacle to typical machine learning approaches, affecting algorithms' convergence to optimum solutions. The complexity and non-convexity of objective functions, which can have several local minima in addition to a global minimum, are at the core of this problem. Points in the parameter space where the objective function approaches a local low are known as local minima, however they are not always the lowest points overall. In order to minimize the objective function, model parameters are frequently

updated iteratively using gradient-based optimization techniques such as gradient descent. Nevertheless, if initialization or parameter updates push these algorithms into areas of the parameter space that match these less-than-ideal solutions, they may become stuck in local minima (*Ghassemi et al., 2020*). Reaching the best possible model performance requires breaking out of such local minima. To address these problems, various metaheuristics algorithms have been employed by researchers with ML algorithms.

## Hybrid approaches using metaheuristics and ML algorithms

In the literature, metaheuristics have been employed by researchers to select features, to optimize parameters, and to train the standard ML algorithms for improving their classification accuracy by avoiding local minima.

Most studies in the literature have used metaheuristic approaches for feature selection problems without focusing on training ML algorithms, to select the most appropriate features among different types of CVD datasets (*Murugesan et al., 2021*; *Muliawan, Rizal & Hadiyoso, 2023*; *Torthi et al., 2024*; *Sampathkumar & Periyasamy, 2024*). These works differ from the proposed work in that metaheuristic methods are used to train ML algorithms while preserving their good qualities. In feature selection problems, metaheuristics help to reduce dimensionality and speed up computation time. One of these studies is the work of *Murugesan et al. (2021)*, where authors created a super learner by fusing three bioinspired algorithms with ANN. Using the methods for BFO (bacterial foraging optimization), KH (krill herd), and CSO (cat swarm optimization), three sets of features were chosen. Using the characteristics chosen by each method, a backpropagation neural network (BPNN) was trained. The Statlog dataset yielded an accuracy of 86.36%, whereas the Cleveland dataset produced an accuracy of 84%.

The study by *Muliawan, Rizal & Hadiyoso (2023)* uses ensemble classifiers with parameter optimization to predict heart disease using a public dataset from the UCI machine learning repository. The dataset includes 13 variables influencing heart disease. Particle swarm optimization (PSO) was used for feature selection and principal component analysis (PCA) for feature extraction. Parameter optimization was applied to machine learning methods like SVM, deep learning, and ensemble classifier. The results showed the highest accuracy in Deep Learning and SVM parameters, with bagging on SVM achieving 83.51% accuracy.

*Torthi et al. (2024)* proposed BAPSO-RF, a Bat algorithm and particle swarm optimization-based Random Forest, to improve heart disease prediction accuracy. Using 270 records and 14 variables from the UCI heart disease dataset, the proposed BAPSO-RF is assessed. The model outperformed other methods like GAPSO-RF, GA, and GA-RBF by using metrics like accuracy, precision, recall, and f1-score values of approximately 98.71%, 98.67%, 98.23%, and 98.45%, respectively.

*Sampathkumar & Periyasamy (2024)* proposed a method using binary particle swarm optimization and attention-based deep network (BPSO-ADN) to extract significant features from a cardiac dataset for improved prediction accuracy. The technique uses BPSO for feature selection and ADN for detailed pattern analysis, with BPSO directed by a fitness evaluation method to find the most suitable subset for heart disease prediction.

In some research works, metaheuristics have been employed by researchers to optimize parameters for improving their classification accuracy (*Nalluri et al., 2017*; *Dhanka & Maini, 2024*). These works differ from the proposed work in that metaheuristic methods are used to train ML algorithms while preserving their good qualities. Metaheuristics are a useful tool for effectively exploring complicated search spaces in hyperparameter optimization issues. This aids in the finding of optimum or nearly optimal configurations for machine learning models. These techniques, which use parallelization, stochastic search, and adaptive exploration to break free from local optima and enhance model performance over a wide range of tasks and datasets, provide adaptable and reliable solutions. One of these studies is the work of *Nalluri et al. (2017)*, where authors diagnosed heart disease using the hybrid system. The classification was done using SVM and MLP classifiers. The parameters were optimized using three evolutionary algorithms: PSO, FA (frefy algorithm), and GSA (gravity search algorithm). Learning rate and momentum were optimized in MLP. Margin was optimized in SVM. Five datasets related to cardiovascular disease were used to verify the system. On the Cleveland dataset, 90.74% on the Statlog dataset, 89.5% on the SPECT dataset, 90.6% on the SPECTF dataset, and 91.4% on the Eric dataset, the system achieved an accuracy of 94.1%.

*Dhanka & Maini (2024)* developed two clever models—HyOPTRF (Model 1) and HyOPTXGBoost Classifier (Model 2)—that were applied to the Statlog HD dataset using both hyper-tuned and default parameters. On Trial (2) the HyOPTRF recorded the highest Accuracy of 92.59% and F1 score 93.75%, while on Trial (33) the HyOPTXGBoost Classifier recorded the highest Accuracy of 96.30% and F1 Score 96.77%. The suggested models were compared to the other models that were already in use and verified using the Stratify Kfold Cross-Validation approach.

In a few research works, metaheuristics have been employed by researchers to train the standard ML algorithms for improving their classification accuracy by avoiding local minima (*Leema, Nehemiah & Kannan, 2016*; *Arabasadi et al., 2017*; *Poornima & Gladis, 2018*; *Shahid & Singh, 2020*; *Al Bataineh & Manacek, 2022*). The resulting algorithms which combine metaheuristics and ML algorithms are referred as hybrid algorithms in the literature. For example, In *Leema, Nehemiah & Kannan (2016)*, the authors aimed to enhance the performance of ANNs by applying a hybrid optimization algorithm. This approach was implemented on three benchmark datasets: Wisconsin Breast Cancer, Pima Indian Diabetes, and Cleveland. The ANN training incorporated a combination of DE and PSO for global search and the backpropagation (BP) algorithm for local search. Prior to constructing the model, the datasets underwent min-max normalization. A 10-fold cross-validation was conducted, and the proposed approach, termed Differential Evolution with Global Information and Back Propagation (DEGI-BP), was compared with DE-BP and PSO-BP. The experiments conducted on the Cleveland dataset demonstrated that the proposed approach outperformed other hybrid optimization algorithms, achieving an accuracy of 86.66%.

A neural network-based approach for diagnosing CAD was proposed by *Arabasadi et al. (2017)*. GA optimized the neural network's weights. The ANN was trained using the backpropagation technique. GA started with an initial population of 100 chromosomes.

The root mean square error, or RMSE, of the untrained ANN was used to determine the chromosomes' fitness value. GA employed the roulette wheel algorithm for selection. With a crossover probability of 1, two-point crossover was employed. Gaussian mutation was used to carry out the mutation. Every gene on a chromosome carried one neural network weight, and every chromosome had all of the neural network's weights. SVM was used to choose features. The system was evaluated on the Z-Alizadeh Sani dataset. The system achieved an accuracy of 93.85%.

A hybrid classifier was presented by *Poornima & Gladis (2018)* to predict cardiac disease. Using the orthogonal local preserving projection (OLPP), features were chosen. The ANN was used to carry out the classification. The neural network consisted of four neurons in the input layer, one hundred neurons in the hidden layer, and five neurons in the output layer. The range of weights for the connections between neurons was between −10 and 10. Levenberg–Marquardt (LM) and group search optimization (GSO) were used to optimize the network by determining the weights. The optimal weights in the network were selected from the two sets of weights that LM and GSO had produced. The results were validated using three datasets: Switzerland, Hungarian, and Cleveland. Using the Cleveland dataset, the accuracy rate of the method was 94%.

*Shahid & Singh (2020)* introduced a pioneering approach, amalgamating PSO with an emotional neural network (EmNN). The performance of this novel approach was compared with a hybrid model named PSO-ANFIS, which integrates an artificial neural network (ANN) with fuzzy logic. The study concentrated on leveraging brain-based emotional learning within EmNNs, renowned for their heightened accuracy. PSO was employed to optimize the proposed neural network. The evaluation encompassed three datasets: Z-Alizadeh Sani, Cleveland, and Statlog. While data preprocessing was omitted, feature selection was conducted, resulting in the selection of 8 features for the Statlog dataset and 7 features for the Cleveland dataset. The achieved accuracy was 84% for the Cleveland dataset and 85.2% for the Statlog dataset.

In a study conducted by *Al Bataineh & Manacek (2022)* utilizing the Cleveland dataset with 13 features and 303 samples, a hybrid algorithm named MLP-PSO was introduced. This algorithm involved replacing missing values with feature-specific mean values and incorporated categorical data encoding and feature scaling techniques. The MLP model was trained using weights and biases optimized through the particle swarm optimization (PSO) algorithm. Performance evaluation was conducted using 5-fold cross-validation, and hyperparameter tuning was executed through the grid search method. Upon comparing the results with ten different machine learning algorithms, the proposed MLP-PSO method demonstrated the highest accuracy, reaching 84.60%. To enhance classification performance, they refined the feature extraction process and training step of a neural network (NN) following the methodology described in *Cherian, Thomas & Venkitachalam (2020)*. Statistical and higher-order statistical features were extracted from the dataset, and principal component analysis (PCA) was performed.

The findings demonstrate that metaheuristic-based ML methods for diagnosing CVD exhibit superior performance over alternative ML techniques in terms of different performance criteria. However, these methods generally require significant time, to improve

regarding low detection rates and effort to achieve high classification performance, which negatively impacts their applicability and effectiveness. To minimize these disadvantages of current hybrid methods, the ML method to be used must first be computationally efficient for large datasets and have simplicity-interpretability in disease diagnosis problems. Afterwards, it is necessary to choose the metaheuristic algorithm that will best overcome the disadvantages of the ML method to be used and will be successful for the train and suitable for the relevant problem (*Naser et al., 2024*).

To eliminate the shortcomings of existing studies and increase the low detection rate in CVD prediction, a pioneering hybrid classifier, denoted as CSA-DE-LR, is introduced in this research endeavor (*Naser et al., 2024*). In this study, the diagnosis of CVD utilizes LR as a classification model because it is computationally efficient for large data sets and simple interpretability in disease diagnosis problems. Then, CSA-DE optimization method employees for model training instead of the gradient descent algorithm to improve LR's classification accuracy by avoiding local minima.

The reasons for choosing CSA-DE as an optimization method can be justified as follows. The CSA is a crucial AIS algorithm that improves the immune system's response to antigens by generating antibodies with greater affinity over time. CSA is popular for optimization tasks because it can search for local and global solutions in the solution space using hyper-mutation and receptor editing processes (*Rahman et al., 2023*). Furthermore, studies have demonstrated that CSA-based techniques outperform other bio-inspired and optimization methods in various contexts (*Haktanirlar Ulutas & Kulturel-Konak, 2011*; *Duru et al., 2022*; *Rahman et al., 2023*). However, traditional CSA techniques may not offer sufficient search capabilities and could benefit from improvement (*Gong, Jiao & Zhang, 2010*; *Zhang et al., 2008*; *Xu et al., 2017*). Therefore, to improve the local search performance of the CSA method, a hybrid optimization method can be developed by using methods that are known to have remarkable search performance, such as the differential evolution (DE) algorithm (*Mostafa et al., 2024*; *Azevedo, Rocha & Pereira, 2024*; *Song et al., 2024*).

As a summary, proposed hybrid classifier CSA-DE-LR combines two optimization algorithms, leveraging their strengths and applying them to LR during training to attain heightened performance. Significantly, it offers three optimization options—based on F1 score, MCC, and MAE—enhancing its adaptability. Additionally, implementing feature selection in this study significantly enhanced the outcomes. By identifying and utilizing the most relevant features, the CSA-DE-LR method achieved remarkable accuracy and efficiency, demonstrating the value of meticulous feature selection in improving diagnostic models for CVD. Employing Bayesian optimization for fine-tuning hyperparameters and utilizing ten-fold cross-validation, CSA-DE-LR demonstrates a notable improvement in diagnostic accuracy across datasets, presenting a substantial advancement in medical diagnostics.

## METHODS

### Logistic regression

Selecting Logistic Regression (LR) is a form of statistical modeling that's especially suitable for situations where the outcome variable is binary, meaning it takes on two possible outcomes. The primary concept behind LR is to model the probability that a given input point belongs to a particular category. This probability estimation is achieved by fitting the data to a logistic curve, hence the name "Logistic Regression."

The training set of features $\{(\vec{x}_1, y_1), \ldots, (\vec{x}_M, y_M)\}$ comprises $M$ instances. Each instance has a feature vector $\vec{x}_i \in R^D$ and the corresponding $y_i$ is the label for each feature vector, which, in this binary classification scenario, can either be 0 or 1.

In mathematical terms, the prediction for a given $\vec{x}_i$ is determined by inputting the weighted sum of its features into the sigmoid function, and Eq. (1). is used to describe the class of Eq. (1).

$$y_i' = \begin{cases} 0, & p_i < 0.5 \\ 1, & p_i \geq 0.5 \end{cases} \tag{1}$$

where $p_i$ is determined by Eq. (2). LR uses the sigmoid function, which outputs a value between 0 and 1 for any input as outlined in Eq. (3). This value can be interpreted as the probability that the input instance belongs to the class labeled as 1.

$$p_i = \sigma(\vec{w}\vec{x}_i) \tag{2}$$

$$\sigma(a) = \frac{1}{1+e^{-a}}. \tag{3}$$

The goal of the learning algorithm is to adjust its internal parameters (typically weights associated with each feature and a bias term) to minimize the difference between its predicted probabilities and the actual outcomes in the training set. This difference is often captured by a cross-entropy cost function given in Eq. (4).

$$J(\vec{w}) = -\sum_{i=1}^{m} y_i \log(p_i) + (1-y_i)\log(1-p_i) \tag{4}$$

### Clonal selection algorithm

The CSA (*de Castro & Von Zuben, 2002*) emulates the adaptive immune system's response to antigenic stimuli. An algorithm, CLONALG, was developed based on the principles of clonal selection and affinity maturation inherent in immune responses. Different adaptations of CLONALG were employed to handle pattern recognition and optimization problems (*de Castro & Von Zuben, 2002*; *Azevedo, Rocha & Pereira, 2024*; *Duru et al., 2022*). For the application of CSA, a version of CLONALG optimized for such tasks was employed. The primary objective of CSA is to identify an antibody with peak affinity (*de Castro & Von Zuben, 2002*; *Rahman et al., 2023*). The key steps of CSA are outlined in Algorithm 1.

---

**Algorithm 1** The Clonal Section Algorithm

---

1: Set up the control parameters: total antibodies ($P$), the highest iteration count, receptor editing rate ($B$), and clonal multiplication coefficient ($\alpha$).
2: Create an initial set of $P$ antibodies.
3: Determine the affinity score for every antibody $Ab$.
4: **for** each iteration **do**
5:     Clone the antibodies $\alpha$ times and calculate the affinity scores for these clones.
6:     **for** each clone $C_i$ **do**
7:         Apply reverse mutation to $C_i$, resulting in the mutated clone $\sigma_i$.
8:         **if** $f(\sigma_i) > f(C_i)$ **then**
9:             $C_i := \sigma_i$
10:        **else**
11:            Carry out pair-wise mutation on $C_i$ to produce $\sigma_i$.
12:            Compute the affinity value of $\sigma_i$.
13:            **if** $f(\sigma_i) > f(C_i)$ **then**
14:                $C_i := \sigma_i$
15:            **else**
16:                $C_i := C_i$
17:            **end if**
18:        **end if**
19:    **end for**
20:    **for** each antibody $Ab_i$ **do**
21:        Choose the clone $C_j$ from $Ab_i$ clones with the topmost affinity
22:        $Ab_i := C_j$
23:    **end for**
24:    Substitute the lowest performing $B$% of antibodies with the newly formed ones.
25: **end for**

---

The set of P antibodies $Ab = \{Ab_1, Ab_2, \ldots, Ab_P\}$ is initially formed randomly as indicated in Eq. (5). This set undergoes enhancement in every cycle through processes like selection, cloning, hyper-mutation, re-selection, and receptor-editing until the highest iteration count is reached (*de Castro & Von Zuben, 2002*). Each antibody $Ab_i = [Ab_{i,1}, Ab_{i,2}, \ldots, Ab_{i,D}] \in R^D$ within the group represents a potential solution. The ultimate goal is to identify an antibody with the utmost affinity score once all cycles have concluded.

$$Ab_{i,j} = lb_j + rand(0, 1) \times (ub_j - lb_j) \tag{5}$$

where $rand(0, 1)$ is a function producing random numbers uniformly spread between 0 and 1. The terms $lb_j$ and $ub_j$ denote the minimum and maximum limits for the $j$th parameter, respectively. After establishing a population of $P$ antibodies, the fitness score for every antibody in the $Ab$ set can be determined as illustrated in Eq. (6).

$$f(Ab_i) = \frac{1}{1 + J(Ab_i)} \tag{6}$$

$f(Ab_i)$ represents the fitness function determining the fitness score of $Ab_i$. Meanwhile, $J(Ab_i)$ serves as the cost function, as depicted in Eq. (4), providing the cost measure of $Ab_i$. The clone count($\alpha_i$) for every chosen antibody $Ab$ may remain consistent (*de Castro & Von Zuben, 2002*). The aggregate count of clones within the clone group $C$ is derived from Eq. (7).

$$|C| = \sum_{i=1}^{n} \alpha_i = \alpha \times n \tag{7}$$

where $\alpha$ represents the clonal multiplication coefficient and is a positive whole number. In this study, every antibody in the group is chosen for cloning ($n = P$), and the quantity of

clones for each selected antibody remains consistent ($\alpha_i = \alpha$) to aid in identifying multiple optimal solutions (*de Castro & Von Zuben, 2002*).

Following the formation of the clone group, the antibodies undergo enhancement *via* hyper-mutation processes, namely inverse mutation and pair-wise mutation. In the inverse mutation method, for each clone $C_i = [C_1, C_2, \ldots, C_D]$, parameters $j$ and $l$ are chosen at random, ensuring $|j - l| > 2$, and the parameters between $j$ and $l$ within $C_i$ are inverted to produce the mutated clone $\sigma_i$. If $\sigma_i$'s affinity surpasses that of $C_i$, it replaces $C_i$. If not, pair-wise mutation is applied to $C_i$. Here, parameters $j$ and $l$ of $C_i$ are randomly selected and swapped. The affinity of $\sigma_i$, resulting from pair-wise mutation, is assessed. If $\sigma_i$'s affinity is superior to that of $C_i$, it takes the place of $C_i$; if not, $C_i$ remains unaltered.

Post hyper-mutation, a re-selection step ensures the antibody population size stays consistent. For each antibody, $Ab_i$, the highest affinity clone among $Ab_i$'s clones is chosen and allocated to $Ab_i$. Concludingly, receptor editing is executed, substituting the least efficient $B$% of antibodies with new ones. The procedures of the CSA optimization approach are detailed in Algorithm 1.

## Differential evolution

The DE algorithm (*Storn & Price, 1997*) operates as a collective method that encompasses processes like crossover, mutation, and selection. Its core mechanism hinges on mutation, which derives from the distinctions between randomly chosen solution pairs within the collective. This algorithm harnesses mutation as an exploration tool and the selection process to guide the exploration towards favorable areas in the solution environment. The DE algorithm also employs a distinctive crossover that may favor parameters from one parent over another. By leveraging attributes from current collective members to formulate trial solutions, the crossover operator adeptly redistributes insights about potent combinations, facilitating a more effective search for optimal solutions. Initially, DE establishes a random set of solution vectors. This set undergoes enhancements through the application of mutation, crossover, and selection processes. In the DE method, every newly generated solution is compared against a mutated one, and the superior of the two emerges victorious. The DE algorithm has captured the attention of scholars in diverse fields and has proven valuable in solving numerous real-world challenges (*Storn & Price, 1997*; *Corne et al., 1999*; *Mostafa et al., 2024*; *Song et al., 2024*; *Azevedo, Rocha & Pereira, 2024*). The essential steps of the DE algorithm are described as follows:

---

**Algorithm 2** Differential Evolution Algorithm

---
1: Initialize Population
2: Evaluation
3: **repeat**
4:     Mutation
5:     Recombination
6:     Evaluation
7:     Selection
8: **until** requirements are met

---

During the mutation process, all of the $M$ parameter vectors are subjected to mutation. This mutation step broadens the exploration area. A mutated solution vector, denoted as $w_i$, is formulated by Eq. (8):

$$\vec{w}_i' = \vec{w}_{i_1} + sf \times (\vec{w}_{i_3} - \vec{w}_{i_2}), \quad 1 \leq i \leq M \tag{8}$$

where $sf$ represents the scaling factor with values from $[0, 1]$, and the solution vectors $i_1$, $i_2$, and $i_3$ are selected randomly and are required to conform to $i_1 \neq i_2 \neq i_3 \neq i$, where i represents the current solution's index. During the crossover procedure, the parent vector merges with the mutated vector, generating a trial vector as given in Eq. (9).

$$\vec{w}_{i,j} = \begin{cases} \vec{w}_{i,j}', & r_j \leq cr \\ \vec{w}_{i,j}, & r_j > cr \end{cases} \tag{9}$$

where $cr$ represents the crossover constant, $r_j$ is a real number picked randomly from the range $[0, 1]$, and $j$ indicates the $j$th element of the related array.

Every solution within the population possesses an equal probability of being chosen as a parent, regardless of its fitness value. After undergoing mutation and crossover processes, the offspring's performance is assessed. Subsequently, a comparison between the offspring and its parent takes place, with the superior entity prevailing. If the parent remains superior, it is preserved in the population.

## Proposed method (CSA-DE-LR)

This study introduces a novel classification methodology, leveraging a hybrid approach that combines CSA and DE to optimize the LR weights for classification tasks. The proposed method, henceforth referred to as CSA-DE-LR, offers three optimization techniques based on different performance metrics: F1 score, MCC, and MAE. These metrics guide the training process to fine-tune the model weights to achieve an optimal balance between precision and generalizability.

---

**Algorithm 3** Proposed CSA-DE-LR classification method

---

**1: Determine the input parameters:** Input matrix $X_{M \times N}$, target $\vec{y}_M$, number of antibodies $P$, population of $P$ antibodies $W_{P \times D}$, percentage of receptor editing $B$, maximum evaluation number $MEN$, lower bound $lb$, upper bound $ub$, number of clones for each antibody $\alpha$, scaling factor $sf$, crossover rate $cr$
**Output:**
  1: $D \leftarrow N + 1$
  2: $W \leftarrow CreateAntibodies(P, D)$
  3: $W' \leftarrow W$
  4: $fit \leftarrow CalculateFitness(W)$
  5: $evaluation\_number \leftarrow 0$
  6: **while** $evaluation\_number < MEN$ **do**
  7:     $Cloning()$
  8:     $LocalSearchViaDE()$
  9:     $Selection()$
 10:     $ReceptorEditing()$
 11:     $FindBestAntibody()$
 12: **end while**
 13: $return \quad \vec{gpar}$                      ▷ return global best antibody params

---

---

**Algorithm 4** Create population of $P$ antibodies

---

1: **procedure** $CreateAntibodies(P, D)$
2:      **for** $i \leftarrow 1 : P$ **do**
3:          **for** $j \leftarrow 1 : D$ **do**
4:              $W[i, j] \leftarrow lb + rand(0, 1) \times (ub - lb)$
5:          **end for**
6:      **end for**
7:      $return \quad W$
8: **end procedure**

---

---

**Algorithm 5** Clone each antibody $\alpha$ times

---

1: **procedure** $Cloning()$
2:      **for** $i \leftarrow 1 : P$ **do**
3:          $C[i \times \alpha : (i+1) \times \alpha, :] \leftarrow W[i, :]$
4:      **end for**
5: **end procedure**

---

---

**Algorithm 6** Local search via DE

---

1: **procedure** $LocalSearchViaDE()$
2:      **for** $i \leftarrow 1 : P \times \alpha$ **do**
3:          $j \leftarrow i // \alpha$                                                      ▷ //, floor division
4:          $\vec{inds} \leftarrow \{x | x \in \mathbb{Z}, 0 \leq x < P, x \neq j\}$
5:          $pars = rand\_choice(\vec{inds}, 3) \times \alpha + randint(0, \alpha, 3)$       ▷ randomly select 3 neighbour clones
6:          $\vec{arr} \leftarrow C[pars[0], :] + sf \times (C[pars[2], :] - C[pars[1], :])$
7:          $\vec{ar} \leftarrow rand(low = 0, high = 1, size = (D))$
8:          $\vec{\rho} \leftarrow \vec{ar} \leq cr$                                          ▷ param to change
9:          $C[i, \vec{\rho}] \leftarrow \vec{arr}[\vec{\rho}]$
10:         $\vec{vec} \leftarrow C[i, \vec{\rho}]$
11:         $\vec{vec}[\vec{vec} < lb] \leftarrow lb$
12:         $\vec{vec}[\vec{vec} > ub] \leftarrow ub$
13:         $C[i, \vec{\rho}] \leftarrow \vec{vec}$
14:     **end for**
15:     $cfit \leftarrow CalculateFitness(C)$
16: **end procedure**

---

---

**Algorithm 7** Selection Phase

---

1: **procedure** $Selection()$
2:      $\vec{cfit}' \leftarrow reshape(\vec{cfit}, size = (P, \alpha))$
3:      $\vec{max_i dxs} \leftarrow argmax(\vec{cfit}', axis = 1)$
4:      $\vec{inds} \leftarrow [0 : 1 : P] \times \alpha$
5:      $\vec{idxs} \leftarrow \vec{inds} + \vec{max_i dxs}$
6:      $\vec{bestIdxs} = \vec{cfit}[\vec{idxs}] > \vec{fit}$
7:      $W[\vec{bestIdxs}, :] = C[\vec{idxs}, :][\vec{bestIdxs}, :]$
8:      $\vec{fit}[\vec{bestIdxs}] = \vec{cfit}[\vec{idxs}][\vec{bestIdxs}]$
9: **end procedure**

---

---

**Algorithm 8** Receptor Editing Phase

---

1: **procedure** $ReceptorEditing()$
2:      $\vec{fIndex} \leftarrow argsort(\vec{fit})$
3:      $n \leftarrow round(P \times B)$
4:      $\vec{worstNindex} \leftarrow \vec{fIndex}[0 : n]$
5:      $newNantibodies \leftarrow CreateAntibodies(n, D)$
6:      $newNfitness \leftarrow CalculateFitness(newNantibodies)$
7:      $W[\vec{worstNindex}, :] = newNantibodies$
8:      $\vec{fit}[\vec{worstNindex}] \leftarrow newNfitness$
9: **end procedure**

---

---

**Algorithm 9** Find best antibody

---
1: **procedure** *FindBestAntibody*()
2:     $index \leftarrow argmax(\vec{fit})$
3:     $gmax \leftarrow \vec{fit}[index]$                                                          ▷ global maximum
4:     $g\vec{par} \leftarrow W[index,:]$                                                           ▷ global params
5: **end procedure**

---

**Algorithm 10** Calculate prediction

---
1: **procedure** *CalculatePrediction*($\phi$)

2:     $w \leftarrow \phi[:,1:]$

3:     $b \leftarrow \phi[:,0]$

4:     $prediction \leftarrow \sigma(X.dot(w^T)+b)$                              ▷ $\sigma$ is sigmoid func

5:     $return \quad prediction$

6: **end procedure**

---

**Algorithm 11** Calculate fitness function using Matthews correlation coefficient

---
1: **procedure** *CalculateFitnessMCC*($\phi$)
2:     $a \leftarrow CalculatePrediction(\phi)$
3:     $p \leftarrow round(a)$                         ▷ round function round elements 1 if elements $\geq 0.5$, otherwise 0
4:     $f \leftarrow MCC(\vec{y_M},p)$
5:     $evaluation\_number \leftarrow evaluation\_number + len(f)$
6:     $return \quad f$
7: **end procedure**

---

**Algorithm 12** Calculate fitness function using F1 score

---
1: **procedure** *CalculateFitnessF1*($\phi$)
2:     $a \leftarrow CalculatePrediction(\phi)$
3:     $p \leftarrow round(a)$                         ▷ round function round elements 1 if elements $\geq 0.5$, otherwise 0
4:     $f \leftarrow F1(\vec{y_M},p)$
5:     $evaluation\_number \leftarrow evaluation\_number + len(f)$
6:     $return \quad f$
7: **end procedure**

---

**Algorithm 13** Calculate fitness function using mean absolute error (MAE)

---
1: **procedure** *CalculateFitnessMAE*($\phi$)
2:     $p \leftarrow CalculatePrediction(\phi)$
3:     $f \leftarrow MAE(\vec{y_M},p)$
4:     $f \leftarrow 1/(f+1)$
5:     $evaluation\_number \leftarrow evaluation\_number + len(f)$
6:     $return \quad f$
7: **end procedure**

---

**Algorithm 14** Calculate Matthews correlation coefficient

---
1: **procedure** *MCC*($actual, predicted$)
2:     $tp \leftarrow sum(predicted * actual, axis = 0)$
3:     $tn \leftarrow sum((1 - predicted) * (1 - actual)), axis = 0)$
4:     $fp \leftarrow sum(predicted, axis = 0) - tp$
5:     $fn \leftarrow sum(actual, axis = 0) - tp$
6:     $mcc \leftarrow (tp * tn - fp * fn)/(tp + fn) * (tp + fp) * (tn + fn) * (tn + fp)$
7:     $return \quad mcc$
8: **end procedure**

---

---

**Algorithm 15** Calculate F1 score

---
1: **procedure** $F1(actual, predicted)$
2:      $tp \leftarrow sum(predicted * actual, axis = 0)$
3:      $fp \leftarrow sum(predicted, axis = 0) - tp$
4:      $fn \leftarrow sum(actual, axis = 0), tp$
5:      $precision \leftarrow tp/(tp + fp)$
6:      $recall \leftarrow tp/(tp + fn)$
7:      $F1 \leftarrow 2 * precision * recall/(precision + recall)$
8:      $return \quad F1$
9: **end procedure**

---

**Algorithm 16** Calculate MAE

---
1: **procedure** $MSE(actual, predicted)$
2:      $mae = mean((actual - predicted)^2, axis = 0)$
3:      $return \quad mae$
4: **end procedure**

---

The CSA-DE-LR method begins by initializing a population of P antibodies, each representing a potential solution to the classification problem. The antibodies undergo cloning and local search procedures *via* DE, followed by a selection phase that favors the most promising candidates. Receptor editing is applied to introduce diversity by replacing a portion of the least-fit antibodies with new candidates. This iterative process continues until the maximum evaluation number (*MEN*) is reached, at which point it returns the best-performing antibody, indicative of the optimal model weights.

The detailed pseudocode for the CSA-DE-LR classification method is presented from Algorithm 3 to Algorithm 11. Algorithm 3 outlines the main procedure, which utilizes sub-procedures defined in Algorithms 4 to Algorithm 11. Each sub-procedure is dedicated to a specific task within the optimization process, including the initialization of the antibody population (Algorithm 4), cloning (Algorithm 5), local search *via* DE (Algorithm 6), selection (Algorithm 7), receptor editing (Algorithm 8), and the identification of the best antibody (Algorithm 9).

A critical phase within this process is detailed in Algorithm 7, the Selection Phase, where the efficacy of each antibody's clone is rigorously evaluated. Every antibody's clones are compared, and the clone with the highest fitness value is identified. If this clone surpasses the original antibody in terms of fitness, it is preferentially selected as a superior solution. This approach ensures that the proposed model continuously evolves towards higher accuracy by adopting the most advantageous traits of each generation.

Following this, Algorithm 8, the Receptor Editing Phase, comes into play. Here, antibodies are ranked based on their fitness values, and the bottom percentile, amounting to PxB antibodies, is identified for replacement. This mechanism introduces strategic diversity to the population by substituting the least-fit antibodies with newly created ones, thus preventing premature convergence and maintaining a robust search within the solution space.

Prediction calculations are performed per Algorithm 10, where the sigmoid function ($\sigma$) is applied to the weighted sum of inputs plus the bias term. The fitness of each antibody is

evaluated using one of the three fitness functions (Algorithms 11 to 13), each corresponding to one of the selected optimization metrics.

Algorithms 14 to 16 encapsulate the calculation of the MCC, F1 score, and MAE, respectively. The MCC computation follows the standard formula involving true positives, true negatives, false positives, and false negatives. Similarly, the F1 score is computed using precision and recall derived from the confusion matrix. The MAE, on the other hand, is inverted to ensure that a lower error results in higher fitness.

By integrating CSA and DE with LR, the proposed CSA-DE-LR method aims to effectively navigate the search space and converge to an optimal set of weights for the logistic regression classifier, thereby enhancing classification accuracy and model robustness.

# EXPERIMENTS

## Datasets

In this study, four well-known datasets were used for empirical analysis: the Cleveland, Statlog, Breast Cancer Wisconsin Original (WBCO), and Breast Cancer Wisconsin Diagnostic (WBCD) datasets. All four datasets are publicly available through the UCI Machine Learning Repository and are commonly used in medical classification research.

- The Cleveland dataset contains 303 instances, each described by 13 attributes. It is used to classify instances as indicative of CAD or representing a healthy state.
- The Statlog dataset consists of 270 instances, also described by 13 attributes. It is similar in structure to the Cleveland dataset and is used to classify the presence of CAD.
- The WBCO dataset comprises 699 instances, each described by nine attributes based on biopsy data. This dataset is used to classify tumors as either benign or malignant.
- The WBCD dataset contains 569 instances, each described by 30 attributes. This dataset is used to classify tumors as benign or malignant.

## Preprocessing

Data preprocessing was crucial to ensure consistent scaling and accurate results. First, any missing values were identified and addressed to maintain data integrity. In the Cleveland dataset, six instances with missing values were removed, and in the WBCO dataset, 16 instances containing missing values were excluded. The Statlog and WBCD datasets did not contain any missing values.

After addressing the missing data, the scaling process was carried out. For the Cleveland, Statlog, and WBCO datasets, the training data for each fold was normalized using the MinMaxScaler, which scales data values to the [0, 1] range. For the WBCD dataset, the StandardScaler was applied to normalize the data by centering and scaling based on the mean and standard deviation. The scalers were first fit and applied to the training data, and subsequently, the transformation was applied to the test data to ensure consistent scaling.

Following the scaling procedures, a 10-fold cross-validation process was employed to evaluate the model's performance. This involved dividing the data into 10 equally sized folds. Each fold was used as a test set while the remaining nine folds formed a training set, allowing the model to be trained and evaluated 10 separate times. The individual

results from each fold were then averaged to provide a more comprehensive and reliable assessment of the model's overall performance.

## Evaluation metrics

In this study, several key metrics were utilized to evaluate and compare the performance of the classification processes. These include:

- **Accuracy (ACC)**: This metric measures the ratio of correctly predicted observations to the total observations and the formula for ACC is provided by Eq. (10).

$$ACC = \frac{TP + TN}{TP + TN + FP + FN} \tag{10}$$

where TP = True Positives, TN = True Negatives, FP = False Positives, FN = False Negatives.

- **F1 score**: The F1 score is a metric that balances precision (the quality of the positives identified) and recall (the ability to find all relevant instances). The F1 score is calculated as:

$$F1 = 2 \times \frac{Precision \times Recall}{Precision + Recall} \tag{11}$$

where Precision = TP/(TP + FP), Recall = TP/(TP + FN). This metric is critical for understanding the model's accuracy in classifying positive cases.

- **Matthews correlation coefficient (MCC)**: This coefficient is a reliable statistical rate which yields a value between -1 and +1. It is especially useful for imbalanced datasets. The formula for MCC is:

$$MCC = \frac{(TP \times TN) - (FP \times FN)}{\sqrt{(TP + FP)(TP + FN)(TN + FP)(TN + FN)}} \tag{12}$$

- **False negative rate (FNR)** and **false positive rate (FPR)**: These rates are crucial for understanding the types of errors made by the model. FNR measures the rate at which positive cases are mistakenly classified as negative, while FPR measures the rate of negative cases incorrectly classified as positive. Their formulas are:

$$FNR = \frac{FN}{TP + FN} \tag{13}$$

$$FPR = \frac{FP}{TN + FP} \tag{14}$$

- **ROC-AUC score**: The receiver operating characteristic (ROC) curve and the area under the curve (AUC) represent the model's ability to distinguish between classes. A score close to 1 indicates perfect classification, while a score around 0.5 is no better than random guessing.

## Hyper-parameter optimization

A crucial aspect of the experimental design involved meticulously optimizing hyperparameters for each classification method, including the CSA-DE-LR, CSA-LR, DE-LR, and other popular machine learning techniques such as decision tree (DT), linear

**Table 1** Hyperparameter ranges and the optimal hyperparameters obtained after 300 iterations for various classifiers on Statlog and Cleveland datasets.

| Classifier | Parameter | Low | High | Statlog (Best) | Cleveland (Best) | WBCD (Best) | WBCO (Best) |
|---|---|---|---|---|---|---|---|
| DT | Min Samples Split | 2 | 100 | 83 | 92 | 3 | 54 |
| | Min Samples Leaf | 1 | 100 | 65 | 76 | 4 | 12 |
| LDA | Shrinkage | 0 | 1 | 0.747 | 0.875 | 0.422 | 0.698 |
| MLP | Learning Rate | $10^{-8}$ | $10^{-1}$ | 0.028 | 0.290 | 0.261 | 0.331 |
| | Number of Hidden Units | 2 | 40 | 6 | 16 | 18 | 25 |
| | Batch Size | 1 | 1024 | 182 | 247 | 501 | 209 |
| | Number of Epochs | 1 | 50 | 22 | 31 | 5 | 33 |
| RF | Number of Trees | 1 | 200 | 75 | 122 | 172 | 168 |
| SVM | C | 0.001 | 1 | 0.014 | 0.577 | 0.953 | 0.772 |
| XGBoost | Eta | 0.1 | 1 | 0.222 | 0.948 | 0.997 | 0.935 |
| | Depth | 1 | 40 | 14 | 15 | 27 | 16 |
| LR | C | $10^{-4}$ | $10^{4}$ | 63576.24 | 19600.49 | 35937.61 | 13519.16 |
| CSA-LR | $lb$ | −64 | −16 | −60.839 | −47.767 | −44.305 | −61.344 |
| | $ub$ | 16 | 64 | 53.250 | 43.126 | 45.543 | 27.328 |
| | $P$ | 10 | 80 | 74 | 61 | 73 | 20 |
| | $\alpha$ | 2 | 6 | 4 | 3 | 4 | 5 |
| | $B$ | 0.05 | 0.2 | 0.050 | 0.162 | 0.104 | 0.194 |
| DE-LR | $lb$ | −64 | −16 | −55.968 | −34.463 | −53.381 | −59.298 |
| | $ub$ | 16 | 64 | 22.412 | 22.809 | 38.460 | 53.334 |
| | $P$ | 10 | 80 | 53 | 29 | 70 | 50 |
| | $sf$ | 0.01 | 2 | 0.940 | 1.790 | 0.116 | 0.257 |
| | $cr$ | 0.01 | 1 | 0.755 | 0.432 | 0.651 | 0.884 |
| CSA-DE-LR | $lb$ | −64 | −16 | −48.353 | −27.010 | −36.035 | −63.765 |
| | $ub$ | 16 | 64 | 19.487 | 24.245 | 29.282 | 32.241 |
| | $P$ | 10 | 80 | 78 | 27 | 37 | 10 |
| | $\alpha$ | 2 | 6 | 4 | 4 | 3 | 5 |
| | $B$ | 0.05 | 0.2 | 0.198 | 0.165 | 0.132 | 0.053 |
| | $sf$ | 0.01 | 2 | 0.670 | 0.070 | 0.167 | 1.610 |
| | $cr$ | 0.01 | 1 | 0.577 | 0.554 | 0.333 | 0.958 |

discriminant analysis (LDA), Logistic Regression (LR), Multi-Layer Perceptron (MLP), Random Forest (RF), XGBoost, and Support Vector Machine (SVM). Table 1 details the hyperparameter ranges and the best values obtained for each classifier on the Statlog, Cleveland, WBCD, and WBCO datasets, achieved after 300 iterations of tuning using the Hyperopt (*Bergstra, Yamins & Cox, 2013*) method.

For the CSA-LR approach, the hyperparameters tuned included the lower and upper bounds (*lb* and *ub*), population size (*P*), number of clones (*α*), and receptor editing rate (*B*). The DE-LR method involved optimizing the lower and upper bounds (*lb* and *ub*), population size (*P*), scaling factor (*sf*), and crossover rate (*cr*). These adjustments ensured that both methods could operate efficiently within their designed optimization frameworks.

**Table 2  Comparative analysis of optimization strategies (F1-Opt, MAE-Opt, and MCC-Opt) of the proposed method on statlog and cleveland datasets using 10-fold cross validation.** Performance metrics: ACC, F1 score, MCC, ROC-AUC Score, FNR, and FPR with standard deviations (Std). The highest values are highlighted in bold.

| Criteria | Statlog | | | Cleveland | | |
|---|---|---|---|---|---|---|
| | F1-Opt | MAE-Opt | MCC-Opt | F1-Opt | MAE-Opt | MCC-Opt |
| ACC ± Std | **88.15 ± 0.039** | 87.78 ± 0.040 | 87.04 ± 0.041 | 86.00 ± 0.053 | 85.67 ± 0.073 | **86.67 ± 0.059** |
| F1 ± Std | **86.73 ± 0.049** | 84.76 ± 0.062 | 83.98 ± 0.062 | 84.44 ± 0.061 | 82.87 ± 0.101 | **84.64 ± 0.066** |
| MCC ± Std | **76.74 ± 0.075** | 75.83 ± 0.081 | 74.46 ± 0.082 | 71.79 ± 0.105 | 71.76 ± 0.145 | **74.32 ± 0.115** |
| ROC-AUC ± Std | **88.42 ± 0.039** | 87.20 ± 0.048 | 86.50 ± 0.048 | 85.67 ± 0.053 | 85.26 ± 0.076 | **86.52 ± 0.056** |
| FNR ± Std | **0.099 ± 0.071** | 0.199 ± 0.109 | 0.199 ± 0.118 | **0.161 ± 0.083** | 0.209 ± 0.144 | 0.191 ± 0.105 |
| FPR ± Std | **0.137 ± 0.046** | 0.057 ± 0.043 | 0.070 ± 0.054 | 0.125 ± 0.065 | 0.085 ± 0.055 | **0.077 ± 0.079** |

In the proposed CSA-DE-LR approach, a combination of hyperparameters was tuned, including the lower and upper bounds ($lb$ and $ub$), population size ($P$), number of clones ($\alpha$), receptor editing rate ($B$), scaling factor ($sf$), and crossover rate ($cr$). The optimization process balanced exploration and exploitation, finding the best parameter settings to maximize performance.

For other classifiers, the DT focused on 'Min Samples Split' and 'Min Samples Leaf' to control tree depth and prevent overfitting. LDA tuned the 'Shrinkage' parameter for improved generalization. LR optimized the 'C' parameter, which controls regularization strength.

In the MLP, hyperparameters like the learning rate, number of hidden units, batch size, and the number of epochs were tuned to discover the best settings. For RF, the number of trees in the forest was the focal point of optimization.

The SVM involved tuning the 'C' parameter, which defines the trade-off between smooth decision boundaries and classifying training points correctly. Finally, XGBoost focused on 'Eta' (learning rate) and 'Depth' (tree depth) to maximize predictive performance.

These comprehensive optimizations *via* Hyperopt were crucial in maximizing each classifier's performance. They allowed an exhaustive search of the hyperparameter space to ensure optimal settings were used across the various datasets.

## PERFORMANCE RESULTS AND DISCUSSION

In the comparative analysis of different optimization strategies applied to the Statlog and Cleveland datasets, distinct patterns of performance emerge, reflecting the diverse characteristics of these datasets. For the Statlog dataset, the F1-Optimization strategy exhibits superior performance, particularly in Accuracy, F1 score, and ROC-AUC metrics, as detailed in Table 2. This indicates a well-balanced approach in terms of precision and recall, which is essential for achieving a harmonious balance in classification tasks where both aspects are equally critical. Such a result suggests that the F1-Optimization is adept at handling the specific nature and distribution of the Statlog dataset, underlining its suitability for similar data types.

On the other hand, the Cleveland dataset presents a different scenario where the MCC-Optimization strategy outperforms others, particularly in the MCC and ROC-AUC metrics.

**Table 3  Comparative performance of various ML methods and the proposed method (CSA-DE-LR) on the Statlog dataset, evaluated using metrics such as ACC, F1 score, MCC, ROC-AUC score, FNR, FPR, and training time in seconds (Time), with results derived from 10-fold cross-validation.** The highest values are highlighted in bold.

| Method | ACC ± Std | F1 ± Std | MCC ± Std | ROC-AUC ± Std | FNR ± Std | FPR ± Std | Time ± Std |
|---|---|---|---|---|---|---|---|
| DT | 80.37 ± 0.057 | 76.67 ± 0.078 | 61.20 ± 0.125 | 80.24 ± 0.062 | 0.254 ± 0.111 | 0.140 ± 0.094 | 0.001 ± 0.000 |
| LDA | 85.18 ± 0.043 | 82.52 ± 0.058 | 70.82 ± 0.084 | 84.95 ± 0.044 | 0.190 ± 0.102 | 0.110 ± 0.079 | 0.001 ± 0.000 |
| MLP | 85.55 ± 0.034 | 82.99 ± 0.049 | 71.17 ± 0.069 | 85.08 ± 0.039 | 0.183 ± 0.107 | 0.114 ± 0.064 | 0.004 ± 0.000 |
| RF | 84.81 ± 0.074 | 81.72 ± 0.091 | 70.00 ± 0.147 | 84.48 ± 0.076 | 0.213 ± 0.122 | 0.096 ± 0.086 | 0.086 ± 0.001 |
| XGBoost | 83.70 ± 0.052 | 81.05 ± 0.068 | 67.87 ± 0.109 | 83.78 ± 0.054 | 0.197 ± 0.093 | 0.127 ± 0.086 | 0.048 ± 0.021 |
| SVM | 83.70 ± 0.041 | 79.71 ± 0.058 | 68.55 ± 0.075 | 83.21 ± 0.042 | 0.260 ± 0.109 | **0.075 ± 0.074** | 0.005 ± 0.000 |
| LR | 83.33 ± 0.041 | 80.50 ± 0.054 | 66.88 ± 0.087 | 83.03 ± 0.044 | 0.206 ± 0.103 | 0.132 ± 0.075 | 0.002 ± 0.001 |
| CSA-LR | 86.30 ± 0.060 | 83.22 ± 0.067 | 72.78 ± 0.108 | 86.29 ± 0.063 | 0.205 ± 0.116 | 0.069 ± 0.061 | 10.36 ± 0.116 |
| DE-LR | 86.30 ± 0.055 | 84.27 ± 0.064 | 72.94 ± 0.099 | 86.84 ± 0.055 | 0.118 ± 0.106 | 0.145 ± 0.098 | 9.175 ± 0.596 |
| CSA-DE-LR | **88.15 ± 0.039** | **86.73 ± 0.049** | **76.74 ± 0.075** | **88.42 ± 0.039** | **0.099 ± 0.071** | 0.137 ± 0.046 | 0.494 ± 0.025 |

This highlights its effectiveness in dealing with potentially imbalanced data structures and its superior capability in distinguishing between classes with higher precision. The superior performance of MCC optimization in this context underscores the importance of choosing an optimization strategy that aligns with the inherent characteristics of the dataset. This divergence in the performance of optimization strategies across the two datasets underscores the necessity of a tailored approach in machine learning applications, considering each dataset's unique aspects. These findings emphasize the significance of context-dependent strategy selection in machine learning endeavors. The variations in performance across the two datasets illustrate that there is no one-size-fits-all approach, and careful consideration must be given to the specific attributes of each dataset to achieve optimal results. This nuanced understanding of the interplay between optimization strategies and dataset characteristics is crucial for developing robust and effective machine-learning models.

The evaluation of classification methods on the Statlog and Cleveland datasets, as depicted in Tables 3 and 4, offers a comprehensive view of the performance of various popular classification techniques. In this context, the proposed CSA-DE-LR method stands out as a novel approach, sparking curiosity and interest. The performance was assessed using metrics such as ACC, F1-score, MCC, ROC-AUC score, FNR, and FPR, with the results obtained through 10-fold cross-validation providing both the mean and standard deviation (Std) for each metric. In the Statlog dataset (Table 3), the CSA-DE-LR method clearly outperformed the other methods across all metrics, achieving the highest ACC, F1, MCC, and ROC-AUC scores. This clear superiority of CSA-DE-LR, particularly in significantly lowering the FNR to 0.099, instills confidence in its effectiveness, making it an efficient choice for the Statlog dataset.

Similarly, for the Cleveland dataset (Table 4), the CSA-DE-LR method again demonstrated superior performance, leading to ACC, F1, MCC, and ROC-AUC scores. While the FNR of CSA-DE-LR was slightly higher compared to XGBoost, it excelled in minimizing the FPR, establishing it as the most effective method for this dataset.

**Table 4** Comparison of several popular classifiers and the proposed method on the Cleveland dataset, measured using metrics like ACC, F1 Score, MCC, ROC-AUC Score, FNR, FPR, and training time in seconds (Time), based on 10-fold cross-validation results. The highest values are highlighted in bold.

| Method | ACC ± Std | F1 ± Std | MCC ± Std | ROC-AUC ± Std | FNR ± Std | FPR ± Std | Time ± Std |
|---|---|---|---|---|---|---|---|
| DT | 81.66 ± 0.060 | 78.19 ± 0.094 | 63.14 ± 0.130 | 80.96 ± 0.067 | 0.238 ± 0.149 | 0.142 ± 0.063 | 0.001 ± 0.000 |
| LDA | 85.33 ± 0.068 | 83.31 ± 0.079 | 71.05 ± 0.132 | 85.04 ± 0.067 | 0.189 ± 0.105 | 0.109 ± 0.092 | 0.001 ± 0.000 |
| MLP | 85.66 ± 0.068 | 83.35 ± 0.088 | 71.78 ± 0.130 | 85.35 ± 0.069 | 0.187 ± 0.136 | 0.104 ± 0.064 | 0.005 ± 0.002 |
| RF | 85.33 ± 0.061 | 83.34 ± 0.067 | 71.05 ± 0.123 | 84.72 ± 0.058 | 0.197 ± 0.100 | 0.107 ± 0.091 | 0.027 ± 0.001 |
| XGBoost | 86.00 ± 0.051 | 84.14 ± 0.055 | 72.43 ± 0.099 | 85.74 ± 0.047 | **0.180 ± 0.087** | 0.104 ± 0.079 | 0.013 ± 0.001 |
| SVM | 83.33 ± 0.066 | 80.40 ± 0.092 | 66.90 ± 0.132 | 82.78 ± 0.068 | 0.228 ± 0.123 | 0.115 ± 0.091 | 0.005 ± 0.000 |
| LR | 83.00 ± 0.078 | 80.78 ± 0.087 | 66.50 ± 0.145 | 82.75 ± 0.075 | 0.205 ± 0.115 | 0.139 ± 0.119 | 0.002 ± 0.001 |
| CSA-LR | 84.00 ± 0.047 | 81.98 ± 0.055 | 68.40 ± 0.088 | 83.86 ± 0.045 | 0.184 ± 0.094 | 0.139 ± 0.082 | 9.107 ± 0.124 |
| DE-LR | 83.33 ± 0.061 | 82.08 ± 0.065 | 67.14 ± 0.123 | 83.31 ± 0.058 | 0.156 ± 0.079 | 0.178 ± 0.099 | 8.755 ± 0.207 |
| CSA-DE-LR | **86.67 ± 0.059** | **84.64 ± 0.066** | **74.32 ± 0.115** | **86.52 ± 0.056** | 0.191 ± 0.105 | **0.077 ± 0.079** | 0.720 ± 0.033 |

The consistently high performance of CSA-DE-LR across both datasets underscores its effectiveness in handling the diverse characteristics of heart disease data. In contrast, other classification methods showed varying levels of efficiency. However, they did not match the balanced performance of CSA-DE-LR, particularly in achieving low false negative rates without a significant increase in false positives. These findings, detailed in Tables 3 and 4, highlight the importance of incorporating advanced optimization techniques like CSA and DE into logistic regression models. CSA-DE-LR's adaptability and accuracy position it as a valuable tool for medical diagnostic applications, offering potential enhancements in diagnostic decision-making processes.

Although the primary focus of the research was on CAD using the Statlog and Cleveland datasets, the performance of the proposed method was also evaluated on the Breast Cancer datasets (WBCO and WBCD) to demonstrate its generalizability and provide additional validation. In evaluating the WBCO and WBCD datasets, the different optimization strategies (F1-Opt, MAE-Opt, and MCC-Opt) produced distinct results, as presented in Table 5. For the WBCD dataset, the MCC-Opt strategy achieved the best performance across all metrics, including ACC (98.93%), F1 (98.57%), MCC (97.76%), and ROC-AUC (98.68%). It also minimized the false negative rate (FNR) at 0.024 and false positive rate (FPR) at 0.003, demonstrating strong classification capabilities. This indicates that the MCC-Opt strategy is highly effective at distinguishing between classes in the WBCD dataset. For the WBCO dataset, the MAE-Opt strategy stood out as the best performer, delivering high scores in ACC (97.94%), F1 (96.93%), MCC (95.49%), and ROC-AUC (98.28%). It also achieved the lowest FNR (0.008) and FPR (0.027), showcasing its effectiveness in classifying breast cancer data accurately. Furthermore, the MAE-Opt strategy required the least amount of time for training on the WBCD dataset, highlighting its computational efficiency.

These findings underscore the importance of using tailored optimization strategies to accommodate the specific characteristics of each dataset. For the WBCO dataset, MAE-Opt provides a strong balance between accuracy and computational efficiency, while

**Table 5  Comparative analysis of optimization strategies (F1-Opt, MAE-Opt, and MCC-Opt) of the proposed method on WBCD and WBCO datasets using 10-fold cross validation.** Performance metrics: ACC, F1, MCC, ROC-AUC, FNR, FPR, and training time (Time) in seconds with standard deviations (Std). The highest values are highlighted in bold.

| | WBCD | | | WBCO | | |
|---|---|---|---|---|---|---|
| **Criteria** | **F1-Opt** | **MAE-Opt** | **MCC-Opt** | **F1-Opt** | **MAE-Opt** | **MCC-Opt** |
| ACC ± Std | 98.21 ± 0.016 | 98.39 ± 0.015 | **98.93 ± 0.010** | 97.65 ± 0.016 | **97.94 ± 0.015** | 97.65 ± 0.013 |
| F1 ± Std | 97.84 ± 0.017 | 97.74 ± 0.022 | **98.57 ± 0.013** | 96.54 ± 0.028 | **96.93 ± 0.026** | 96.64 ± 0.020 |
| MCC ± Std | 95.95 ± 0.035 | 96.52 ± 0.033 | **97.76 ± 0.019** | 94.88 ± 0.037 | **95.49 ± 0.035** | 94.92 ± 0.029 |
| ROC-AUC ± Std | 98.08 ± 0.016 | 97.98 ± 0.019 | **98.68 ± 0.015** | 98.05 ± 0.013 | **98.28 ± 0.012** | 97.90 ± 0.011 |
| FNR ± Std | 0.036 ± 0.034 | 0.035 ± 0.033 | **0.024 ± 0.027** | **0.008 ± 0.016** | **0.008 ± 0.016** | 0.015 ± 0.019 |
| FPR ± Std | 0.003 ± 0.008 | 0.006 ± 0.011 | **0.003 ± 0.007** | 0.031 ± 0.024 | **0.027 ± 0.023** | 0.027 ± 0.024 |
| Time ± Std | 1.208 ± 0.055 | **1.028 ± 0.035** | 1.260 ± 0.056 | **1.242 ± 0.042** | 1.366 ± 0.054 | 1.674 ± 0.062 |

**Table 6  Comparison of the proposed method CSA-DE-LR with LR, CSA-LR, DE-LR, and several popular classifiers on the WBCD dataset, measured using metrics like ACC, F1 score, MCC, ROC-AUC score, FNR, FPR, and training time in seconds (Time), based on 10-fold cross-validation results.** The highest values are highlighted in bold.

| Method | ACC ± Std | F1 ± Std | MCC ± Std | ROC-AUC ± Std | FNR ± Std | FPR ± Std | Time ± Std |
|---|---|---|---|---|---|---|---|
| DT | 93.92 ± 0.026 | 91.81 ± 0.032 | 87.40 ± 0.051 | 93.17 ± 0.031 | 0.105 ± 0.068 | 0.032 ± 0.034 | 0.004 ± 0.001 |
| LDA | 96.07 ± 0.021 | 94.46 ± 0.028 | 91.79 ± 0.040 | 94.97 ± 0.025 | 0.098 ± 0.051 | **0.003 ± 0.007** | **0.001 ± 0.000** |
| MLP | 98.04 ± 0.017 | 97.33 ± 0.023 | 95.88 ± 0.035 | 97.82 ± 0.021 | 0.032 ± 0.043 | 0.011 ± 0.019 | 0.011 ± 0.005 |
| RF | 96.79 ± 0.016 | 95.64 ± 0.023 | 93.30 ± 0.033 | 96.49 ± 0.021 | 0.047 ± 0.053 | 0.023 ± 0.029 | 0.150 ± 0.005 |
| XGBoost | 97.50 ± 0.016 | 96.51 ± 0.024 | 94.68 ± 0.035 | 96.92 ± 0.022 | 0.053 ± 0.046 | 0.009 ± 0.013 | 0.026 ± 0.003 |
| SVM | 97.32 ± 0.022 | 96.27 ± 0.029 | 94.21 ± 0.045 | 96.78 ± 0.024 | 0.053 ± 0.036 | 0.011 ± 0.014 | 0.011 ± 0.000 |
| LR | 96.25 ± 0.036 | 95.03 ± 0.044 | 92.05 ± 0.074 | 95.94 ± 0.036 | 0.052 ± 0.042 | 0.029 ± 0.039 | 0.016 ± 0.011 |
| CSA-LR | 97.67 ± 0.022 | 96.82 ± 0.033 | 95.03 ± 0.050 | 97.33 ± 0.027 | 0.038 ± 0.045 | 0.014 ± 0.019 | 16.89 ± 0.073 |
| DE-LR | 98.03 ± 0.014 | 97.23 ± 0.022 | 95.77 ± 0.032 | 97.50 ± 0.018 | 0.044 ± 0.034 | 0.005 ± 0.011 | 19.59 ± 0.320 |
| CSA-DE-LR | **98.93 ± 0.010** | **98.57 ± 0.013** | **97.76 ± 0.019** | **98.68 ± 0.015** | **0.024 ± 0.027** | **0.003 ± 0.007** | 1.260 ± 0.056 |

MCC-Opt delivers the best classification performance for the WBCD dataset. Ultimately, this comparative analysis demonstrates that the appropriate optimization strategy can significantly impact the performance of a machine learning model, emphasizing the need for a context-driven approach.

In Table 6, the performance of the proposed CSA-DE-LR model on the WBCD dataset shows clear superiority over other classifiers. The model achieves exceptional results across key metrics, with an accuracy of 98.93% and an F1 score of 98.57%, demonstrating its ability to accurately classify the benign and malignant classes. The MCC and ROC-AUC scores of 97.76% and 98.68%, respectively, highlight its robust predictive power and ability to differentiate between the classes. Moreover, with an FNR of 0.024 and an FPR of 0.003, the model minimizes errors and reduces the risk of incorrect classification. Despite the high accuracy, it maintains a training time of 1.26 s, showcasing its efficiency.

Similarly, Table 7 provides insights into the performance of CSA-DE-LR and other classifiers on the WBCO dataset. The proposed model achieves an accuracy of 97.94% and an F1 score of 96.93%, emphasizing its classification prowess. The MCC of 95.49% and ROC-AUC of 98.28% reflect the model's strong ability to correctly identify the two

**Table 7  Comparison of the proposed method CSA-DE-LR with LR, CSA-LR, DE-LR, and several popular classifiers on the WBCO dataset, measured using metrics like ACC, F1 score, MCC, ROC-AUC score, FNR, FPR, and training time in seconds (Time), based on 10-fold cross-validation results.** The highest values are highlighted in bold.

| Method | ACC ± Std | F1 ± Std | MCC ± Std | ROC-AUC ± Std | FNR ± Std | FPR ± Std | Time ± Std |
|---|---|---|---|---|---|---|---|
| DT | 95.00 ± 0.029 | 92.79 ± 0.042 | 89.12 ± 0.061 | 94.82 ± 0.026 | 0.059 ± 0.033 | 0.045 ± 0.044 | **0.001 ± 0.000** |
| LDA | 96.18 ± 0.027 | 94.62 ± 0.035 | 91.76 ± 0.057 | 95.57 ± 0.030 | 0.070 ± 0.049 | **0.018 ± 0.023** | 0.001 ± 0.000 |
| MLP | 97.65 ± 0.015 | 96.60 ± 0.022 | 94.86 ± 0.033 | 97.75 ± 0.015 | 0.020 ± 0.027 | 0.025 ± 0.021 | 0.007 ± 0.004 |
| RF | 97.50 ± 0.015 | 96.32 ± 0.025 | 94.53 ± 0.034 | 97.72 ± 0.014 | 0.019 ± 0.026 | 0.026 ± 0.023 | 0.017 ± 0.000 |
| XGBoost | 96.91 ± 0.019 | 95.54 ± 0.028 | 93.25 ± 0.041 | 97.03 ± 0.020 | 0.031 ± 0.033 | 0.029 ± 0.022 | 0.036 ± 0.004 |
| SVM | 97.21 ± 0.021 | 95.96 ± 0.032 | 93.87 ± 0.047 | 97.37 ± 0.021 | 0.024 ± 0.026 | 0.029 ± 0.024 | 0.006 ± 0.000 |
| LR | 96.62 ± 0.023 | 95.26 ± 0.030 | 92.66 ± 0.048 | 96.36 ± 0.025 | 0.048 ± 0.039 | 0.025 ± 0.022 | 0.002 ± 0.002 |
| CSA-LR | 97.35 ± 0.021 | 96.10 ± 0.033 | 94.24 ± 0.047 | 97.40 ± 0.022 | 0.027 ± 0.040 | 0.024 ± 0.028 | 11.60 ± 0.098 |
| DE-LR | 97.35 ± 0.021 | 96.15 ± 0.033 | 94.27 ± 0.046 | 97.75 ± 0.018 | 0.011 ± 0.017 | 0.033 ± 0.029 | 9.600 ± 0.085 |
| CSA-DE-LR | **97.94 ± 0.015** | **96.93 ± 0.026** | **95.49 ± 0.034** | **98.28 ± 0.011** | **0.007 ± 0.015** | 0.026 ± 0.023 | 1.366 ± 0.054 |

classes. Furthermore, a low FNR of 0.007 and an FPR of 0.026 underscore its reliability in minimizing classification errors. Despite its comprehensive performance, CSA-DE-LR maintains an efficient training time of 1.366 s.

Overall, the results from Tables 3, 4, 6 and 7 highlight that the CSA-DE-LR model offers substantial improvements over other methods, effectively combining the strengths of CSA and DE optimization techniques with logistic regression. Its consistently high performance across diverse metrics ensures reliable and accurate classification, making it a valuable tool for medical diagnostics.

In this study, Wilcoxon signed-rank tests were conducted to evaluate the statistical significance of performance differences between CSA-DE-LR and the other classifiers. Each classifier was tested 30 times across four datasets using the hyperparameters that had previously provided the best performance. The performance results were calculated using 10-fold cross-validation, and the final results are expressed as averages for Accuracy (ACC), F1 score (F1), and Matthews correlation coefficient (MCC). Table 8 demonstrates a consistent and statistically significant difference in classification performance between the proposed CSA-DE-LR method and other classifiers across four datasets. This conclusion is drawn from the Wilcoxon signed-rank tests, which consistently returned very low $p$-values, generally considered significant at less than 0.05, across all three evaluation metrics. These low $p$-values indicate that the performance differences between CSA-DE-LR and the other classifiers are statistically significant and unlikely to occur due to random chance.

In the Cleveland dataset, CSA-DE-LR's performance showed statistically significant improvements compared to other classifiers, with $p$-values suggesting a meaningful difference. In the Statlog dataset, CSA-DE-LR again significantly outperformed other methods, with the Wilcoxon tests confirming substantial differences in performance. Similar trends were observed in the WBCD dataset, where CSA-DE-LR demonstrated its effectiveness in all three metrics, consistently outperforming other classifiers. Finally, in the WBCO dataset, CSA-DE-LR maintained its advantage, with significant improvements across all evaluation metrics.

**Table 8** Wilcoxon test results indicating *p*-values for comparisons between CSA-DE-LR and other classifiers across multiple datasets.

| | Cleveland | | | Statlog | | | WBCD | | | WBCO | | |
|---|---|---|---|---|---|---|---|---|---|---|---|---|
| Classifier | ACC | F1 | MCC | ACC | F1 | MCC | ACC | F1 | MCC | ACC | F1 | MCC |
| LR | 1.86e−09 | 1.86e−09 | 1.86e−09 | 1.86e−09 | 1.86e−09 | 1.86e−09 | 1.86e−09 | 1.86e−09 | 1.86e−09 | 1.86e−09 | 1.86e−09 | 1.86e−09 |
| CSA-LR | 1.86e−09 | 1.30e−08 | 1.86e−09 | 1.86e−09 | 1.86e−09 | 1.86e−09 | 1.86e−09 | 1.86e−09 | 1.86e−09 | 1.86e−09 | 1.86e−09 | 1.86e−09 |
| DE-LR | 1.86e−09 | 3.73e−09 | 1.86e−09 | 2.51e−06 | 1.86e−09 | 3.73e−09 | 9.76e−06 | 8.01e−08 | 8.01e−08 | 2.53e−06 | 3.73e−09 | 3.73e−09 |
| MLP | 9.31e−09 | 1.30e−08 | 9.31e−09 | 1.86e−09 | 1.86e−09 | 1.86e−09 | 1.86e−09 | 1.86e−09 | 1.86e−09 | 1.86e−09 | 1.86e−09 | 1.86e−09 |
| RF | 1.86e−09 | 1.86e−09 | 1.86e−09 | 1.86e−09 | 1.86e−09 | 1.86e−09 | 1.86e−09 | 1.86e−09 | 1.86e−09 | 1.86e−09 | 1.86e−09 | 1.86e−09 |
| XGBoost | 1.86e−09 | 1.86e−09 | 1.86e−09 | 1.86e−09 | 1.86e−09 | 1.86e−09 | 1.86e−09 | 1.86e−09 | 1.86e−09 | 1.86e−09 | 1.86e−09 | 1.86e−09 |
| DT | 1.86e−09 | 1.86e−09 | 1.86e−09 | 1.86e−09 | 1.86e−09 | 1.86e−09 | 1.86e−09 | 1.86e−09 | 1.86e−09 | 1.86e−09 | 1.86e−09 | 1.86e−09 |
| SVC | 1.86e−09 | 1.86e−09 | 1.86e−09 | 1.86e−09 | 1.86e−09 | 1.86e−09 | 1.86e−09 | 1.86e−09 | 1.86e−09 | 1.86e−09 | 1.86e−09 | 1.86e−09 |
| LDA | 1.86e−09 | 1.86e−09 | 1.86e−09 | 1.86e−09 | 1.86e−09 | 1.86e−09 | 1.86e−09 | 1.86e−09 | 1.86e−09 | 1.86e−09 | 1.86e−09 | 1.86e−09 |

These results indicate that the proposed CSA-DE-LR method consistently provides superior classification performance over other classifiers across various data environments. The low *p*-values reflect statistically significant differences, implying that CSA-DE-LR is a reliable and effective classification method that generally outperforms other existing classifiers.

While the proposed method demonstrates reliable and robust classification performance with its various optimization options, there is still significant potential for further enhancing its classification capabilities on the Statlog and Cleveland datasets. In this regard, a detailed examination of the models that produce the best results with F1-Opt and MCC-Opt on the Statlog and Cleveland datasets and analyzing their respective weights could offer invaluable insights for model refinement and feature selection. Figure 1 illustrates the average feature weights derived from a 10-fold cross-validation of the best-performing models for both datasets. The comparative analysis presented in Fig. 1 highlights the predictive power of certain clinical variables in the context of heart disease. Notably, features such as 'cp' (chest pain type), 'thal' (thalassemia), and 'ca' (number of significant vessels colored by fluoroscopy) demonstrate substantial positive weights across both datasets, underscoring their critical importance in predicting cardiovascular events. These features have been consistently acknowledged as crucial factors in diagnosing heart disease, irrespective of patient cohort or dataset characteristics. Conversely, 'thalach' exhibits a significant negative weight, suggesting its inverse correlation with the target variable.

However, some features exhibit contrasting weights between the two datasets, which is particularly intriguing. In Statlog, 'age' shows a positive weight, suggesting a direct correlation with the presence of heart disease, whereas in Cleveland, it has a negative weight, which could indicate a less straightforward or even inverse relationship in that specific patient population. Similarly, 'sex' presents a positive weight in Statlog, while it carries a lesser weight in Cleveland. These discrepancies could be due to demographic differences in the datasets or varying patterns of disease presentation between the groups.

Furthermore, 'trestbps' (resting blood pressure) shows a notable difference, with a high positive weight in Statlog but a negative one in Cleveland. This suggests that the same clinical measurement can have a different prognostic value depending on the dataset,

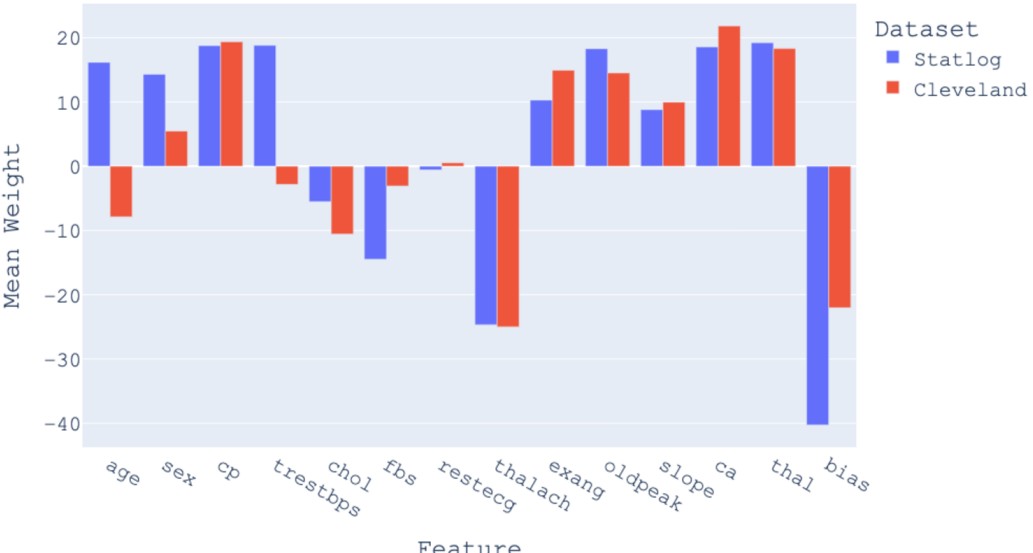

**Figure 1** **Mean weight of each feature for the Statlog and Cleveland datasets.**

potentially influenced by the underlying distribution of the feature, its interaction with other variables, or population-specific health trends.

Additionally, 'restecg' (resting electrocardiographic results) exhibits a weight close to zero in both datasets, implying its limited predictive value in the context of these datasets. This observation aligns with the principle of parsimony, suggesting that the model complexity can be reduced without significant loss of information by omitting this variable. Simplifying the model this way could improve its generalizability and interpretability, making it more accessible for clinical use.

The negative weights for features such as 'chol' (serum cholesterol) and 'fbs' (fasting blood sugar) in both datasets challenge common assumptions about the role of these factors in heart disease, prompting a re-evaluation of their predictive significance. This could reflect the multifactorial nature of heart disease, where the relevance of certain risk factors may be diminished or outweighed by others in specific populations.

In conclusion, this analysis underlines the necessity of dataset-specific model tuning and the careful consideration of feature selection based on their differential impact across datasets. It highlights the importance of context in developing predictive models for heart disease and advocates for a nuanced approach to understanding the contribution of each clinical variable.

Nonetheless, inconsistencies between equivalent features across the Statlog and Cleveland datasets necessitate a more nuanced examination. Accordingly, Figs. 2 and 3 illustrate the fold-specific weights for each feature within the Statlog and Cleveland datasets, respectively. Under normal circumstances, one would anticipate fold weights for the same attribute to exhibit similar directionality and closely clustered values. As depicted

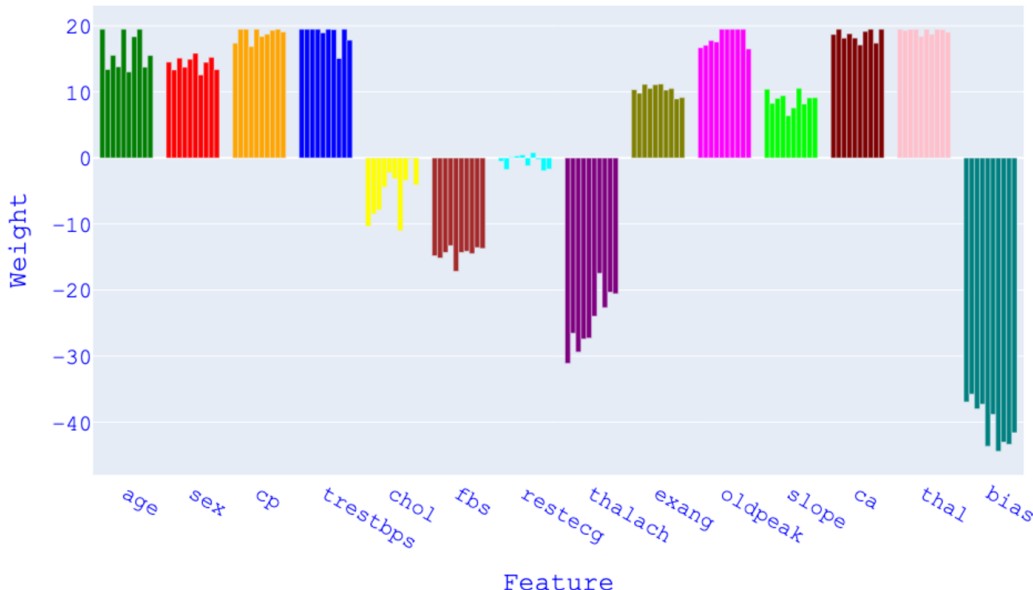

**Figure 2** Fold-specific weights for each feature of the best-performing model on the Statlog dataset.

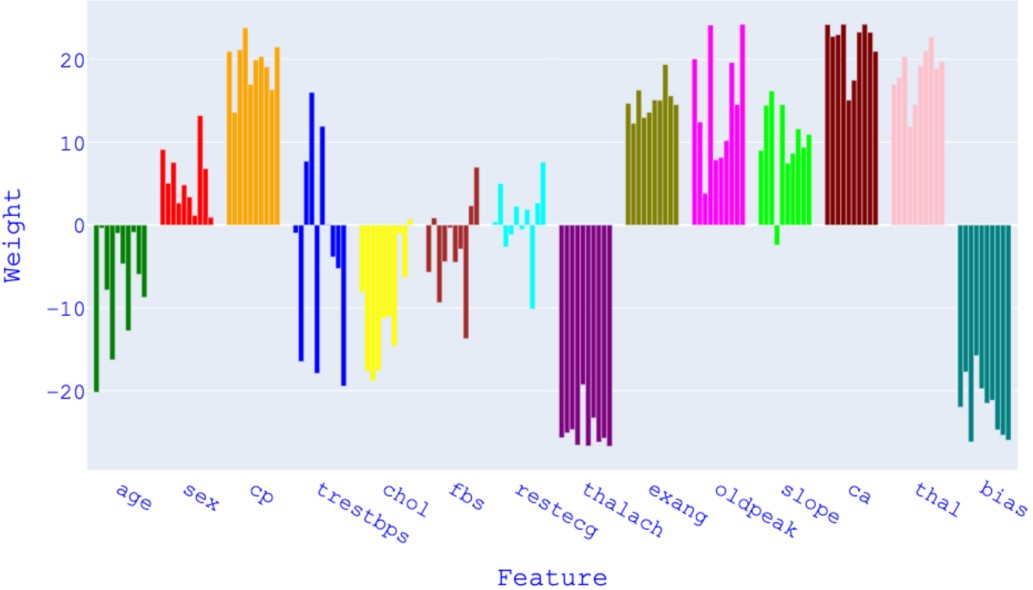

**Figure 3** Fold-specific weights for each feature of the best-performing model on the Cleveland dataset.

in Fig. 2, such consistency is largely maintained for the Statlog dataset, with the notable exception of the 'restecg' attribute.

Prompted by these observations, a methodical reevaluation was undertaken by selectively omitting the 'restecg' feature from the Statlog dataset and the 'trestbps', 'fbs', and 'restecg' features from the Cleveland dataset. This was undertaken to discern the consequential

**Table 9  Enhancing diagnostic performance: a comparative analysis of the CSA-DE-LR method with feature selection on Cleveland and Statlog datasets.**

| Criteria | Statlog (F1-Opt) | Cleveland (MCC-Opt) |
|---|---|---|
| ACC ± Std | 88.15 ± 0.027 | 88.00 ± 0.049 |
| F1 ± Std | 86.86 ± 0.037 | 86.08 ± 0.059 |
| MCC ± Std | 76.63 ± 0.054 | 76.17 ± 0.100 |
| ROC-AUC ± Std | 88.44 ± 0.027 | 87.54 ± 0.049 |
| FNR ± Std | 0.099 ± 0.071 | 0.175 ± 0.089 |
| FPR ± Std | 0.132 ± 0.051 | 0.074 ± 0.050 |

effects on the classification performance and to understand whether excluding these variables would lead to a model with improved predictive consistency and generalizability. Here, this study aims to enhance the model's robustness by eliminating variables that exhibit high variability and do not consistently contribute to predictive accuracy. This strategy is particularly pertinent in the context of medical diagnostics, where the interpretability and reliability of a predictive model are paramount.

Following this strategic reevaluation and feature omission, the performance of the optimized models was tested: CSA-DE-LR with F1-Opt for the Statlog dataset and CSA-DE-LR with MCC-Opt for the Cleveland dataset. These models were executed using the same hyperparameter settings that previously yielded optimal results. The outcomes of this process are detailed in Table 9. Applying these models with refined feature sets aimed to further probe into the effectiveness of the feature selection approach. This step was crucial in determining the impact of feature exclusion on the overall model performance, mainly focusing on predictive accuracy and consistency.

Table 9 presents the best results after implementing feature selection, directly comparing initial models. This comparison not only underscores the tangible benefits of feature selection in enhancing the performance of CSA-DE-LR but also highlights its practical implications. The improvements in accuracy, F1 score, MCC, and ROC-AUC for both datasets illustrate the method's heightened ability to effectively discern and classify cases post feature selection. These enhancements are particularly significant in medical diagnostic applications where precision is paramount, demonstrating the real-world impact of this research. Furthermore, the stability observed in FNR and FPR rates across both datasets post feature selection reaffirms the reliability of the CSA-DE-LR method, providing further assurance of its practicality and usefulness.

In essence, the results in Table 9 validate the efficacy of feature selection in this proposed method and highlight its critical role in achieving a balance between model complexity and performance. This balance is essential in ensuring that the model remains both accurate and interpretable, a vital consideration in the field of medical diagnostics.

Finally, the proposed method is compared with outcomes from previous studies. While numerous studies have utilized the Statlog and Cleveland datasets, only a few have employed the 10-fold cross-validation technique. Studies with similar preprocessing procedures that also applied 10-fold cross-validation were selected to ensure a fair comparison for reasons such as some studies: (i) using the highest value obtained after cross-validation instead of

**Table 10  A historical comparison of CSA-DE-LR performance against previous studies on Cleveland and Statlog heart disease datasets.** The highest values are highlighted in bold.

| Dataset | Method | ACC (%) | F1 (%) | K-Fold CV | Article |
|---------|--------|---------|--------|-----------|---------|
| Cleveland | NN-DEGI-BP | 86.66 | – | 10 | *Leema, Nehemiah & Kannan (2016)* |
| | MLP | 82.50 | 83.80 | 10 | *Kolukısa et al. (2019)* |
| | Ensemble | 83.43 | 81.10 | 10 | *Kolukısa et al. (2020)* |
| | PSO-EmNN | 84 | 82.29 | 10 | *Shahid & Singh (2020)* |
| | MLP-PSO | 84.60 | 84.40 | 5 | *Al Bataineh & Manacek (2022)* |
| | MGOHBO-KELM | 82.22 | – | 10 | *Shan et al. (2022)* |
| | MLP | 85.47 | 83.90 | 10 | *Kolukisa & Bakir-Gungor (2023)* |
| | CSA-DE-LR | **88.00** | **86.08** | 10 | **Proposed method (2024)** |
| Statlog | PSO-EmNN | 85.20 | 84 | 10 | *Shahid & Singh (2020)* |
| | MGOHBO-KELM | 81.85 | – | 10 | *Shan et al. (2022)* |
| | MLP | 85.55 | 85.30 | 10 | *Kolukisa & Bakir-Gungor (2023)* |
| | LR | 85.2 | – | 10 | *Dhanka, Bhardwaj & Maini (2023)* |
| | XGBoost | 81.5 | – | 10 | *Dhanka, Bhardwaj & Maini (2023)* |
| | CSA-DE-LR | **88.15** | **86.86** | 10 | **Proposed method (2024)** |

the mean value of all results (*Nalluri et al., 2017*), (ii) showing resampling results (*Dhanka & Maini, 2024*), and (iii) putting the results obtained on the train set instead of the test set. The effectiveness of the CSA-DE-LR method is not just underscored in a comparative analysis with these studies, as detailed in Table 10, but it also shines through, highlighting its novelty and superiority. This comparison, focusing on both the Cleveland and Statlog datasets, reveals the significant advancements in classification performance achieved by the CSA-DE-LR method, sparking excitement about its potential in the field of medical diagnostics.

For the Cleveland dataset, CSA-DE-LR, optimized through 10-fold cross-validation, demonstrates a notable improvement in both ACC and F1 scores, achieving 88.0% and 86.08%, respectively. This surpasses the results of previous methods like NN-DEGI-BP (2016), various MLP implementations (2019–2023), Ensemble (2020), PSO-EmNN (2020), and MGOHBO-KELM (2022). The improvement is particularly evident when compared with the most recent MLP method in 2023, underscoring the advancements made by CSA-DE-LR in classification accuracy and precision.

Similarly, for the Statlog dataset, CSA-DE-LR outperforms the other methods listed, including PSO-EmNN (2020), MGOHBO-KELM (2022), MLP (2023), LR (2023), and XGBoost (2023). With an ACC of 88.15% and an F1 score of 86.86%, the CSA-DE-LR method demonstrates its superiority in this dataset, further validating the proposed approach's efficacy.

These results, achieved in 2024, position CSA-DE-LR as a leading method in heart disease classification. This method may identify instances of heart disease with lower error rates, as evidenced by its high accuracy, F1-score, and ROC-AUC. Low false negative and false positive rates are attained by CSA-DE-LR, which reduces misdiagnoses and reduces needless therapies while also improving patient outcomes. By providing insights into

important clinical aspects, the feature weights analysis enables more focused and effective diagnostic procedures. These elements, along with the method's resilience and flexibility, imply that CSA-DE-LR can be an effective tool in medical contexts by speeding diagnostic procedures and optimizing resource allocation, hence lowering the cost of healthcare.

### Ethical implications of deploying ML models in healthcare

Healthcare professionals may greatly benefit from ML models, which have the potential to improve patient outcomes and clinical decision-making. To guarantee the proper and ethical use of predictive technology, it is necessary to address important ethical concerns in conjunction with these breakthroughs (*Chotrani, 2021*). Securing patient privacy and confidentiality is one of the ethical issues with machine learning in healthcare (*Manoharan et al., 2023*). Data protection laws were followed, and steps were taken to anonymize patient information in order to reduce privacy concerns. In this work, a cardiovascular disease prediction model was trained using publicly accessible healthcare datasets, including Cleveland and Statlog. These datasets frequently have usage and sharing agreements that the researchers agreed to abide by. These policies may include limitations on redistribution, citation requirements, and ethical considerations.

The possibility of biases in the machine learning model either from the training data or the algorithms themselves is a further ethical concern. Biases can manifest in various forms, including demographic, socioeconomic, or access-related biases (*Mehrabi et al., 2021*). This work included rigorous data pretreatment methods to reduce biases. Addressing biases in the model aims to ensure fairness and equity in predictive healthcare analytics. In this work, we emphasize the significance of human supervision in addition to machine learning forecasts for cardiovascular disorders. Although our models provide insightful information, they should not be used as a substitute for physicians but rather as decision support tools (*Bankins, 2021*). In order to guarantee the proper implementation of models and patient-centered care, collaboration between data scientists, healthcare professionals, and ethicists is essential. Additionally, this work supports accountability and transparency, which are essential values in the moral application of machine learning algorithms (*Hosain et al., 2023*). This involves transparent reporting of performance metrics, documentation of decisions, and addressing errors, fostering trust in the proposed predictive model for clinical practice.

## CONCLUSIONS

This study introduced CSA-DE-LR, a novel hybrid classification method that integrates the clonal selection algorithm (CSA) and differential evolution (DE) to enhance diagnostic accuracy for cardiovascular diseases (CVD). The empirical analysis of the Cleveland and Statlog datasets showed that CSA-DE-LR outperforms current state-of-the-art machine learning methods in classification accuracy and balanced performance. Multiple optimization techniques, including the F1 score, Matthews correlation coefficient (MCC), and mean absolute error (MAE), highlight the method's adaptability across different scenarios.

One of the notable advantages of CSA-DE-LR is its high accuracy and balanced performance, which is consistent across various metrics, indicating robustness. The method's adaptability to specific datasets through customized feature selection enhances its applicability and generalizability in varied contexts. Additionally, incorporating diverse optimization approaches offers the flexibility to cater to different needs and circumstances.

However, the study also acknowledges certain limitations. The computational complexity of the hybrid approach can be higher, especially with larger datasets. The complex nature of these hybrid models could pose challenges in terms of interpretability, particularly in clinical settings.

Future research should consider testing CSA-DE-LR on various medical datasets to validate its generalizability and applicability further. Exploring how adaptable this method is to different medical conditions and diseases could provide broader insights. Additionally, there is scope for enhancing the computational efficiency of the hybrid model and improving its interpretability. Investigating the feasibility of implementing CSA-DE-LR in clinical settings for real-time diagnostic tools could be a significant step forward. Lastly, combining the model with deep learning approaches might enhance its performance and broaden its application spectrum.

In summary, CSA-DE-LR emerges as a promising method that overcomes some limitations of traditional machine learning and metaheuristic approaches to diagnosing CVD. Its potential to improve diagnostic processes in the medical field is substantial, but it requires ongoing exploration and refinement for broader adoption in clinical applications.

### Funding
The authors received no funding for this work.

### Competing Interests
The authors declare there are no competing interests.

### Author Contributions
- Beyhan Adanur Dedeturk conceived and designed the experiments, performed the experiments, analyzed the data, prepared figures and/or tables, and approved the final draft.
- Bilge Kagan Dedeturk conceived and designed the experiments, performed the experiments, analyzed the data, performed the computation work, prepared figures and/or tables, and approved the final draft.
- Burcu Bakir-Gungor conceived and designed the experiments, authored or reviewed drafts of the article, and approved the final draft.

### Data Availability
The data and code files are available in the Supplemental Files.

## Supplemental Information

Supplemental information for this article can be found online at http://dx.doi.org/10.7717/peerj-cs.2197#supplemental-information.

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
