# Peer review of "CSA-DE-LR: enhancing cardiovascular disease diagnosis with a novel hybrid machine learning approach"

_PeerJ Computer Science, doi:10.7717/peerj-cs.2197_

## Round 0.1 · original submission · Major Revisions

Based on the referee reports, I recommend a major revision of the manuscript. The author should improve the manuscript, taking carefully into account the comments of the reviewers in the reports and resubmit the manuscript.

**Language Note:** The review process has identified that the English language must be improved. PeerJ can provide language editing services - please contact us at [email protected] for pricing (be sure to provide your manuscript number and title). Alternatively, you should make your own arrangements to improve the language quality and provide details in your response letter. – PeerJ Staff

Reviewer 1 ·

Basic reporting

1. The usage of English throughout the paper to be checked and also to be improved.
2. The literature survey need to be enhanced with the papers that deals with hybrid architectures
3. The novelty of the paper should be emphasized. It seems that whether the conjunction of CSA-DE-LR is alone novel.
4. Usage of "we", "us" and "our" are to be avoided in the paper.

Experimental design

1. The dataset been taken from the two standard sources. Why dont the real time data set are not attempted?
2. How the accuracy can be compared with other algorithms while the dataset and the procedures are different?
3. The testing of the model postulated is not well addressed.

Validity of the findings

The findings are not validated.

Additional comments

Nil

·

Basic reporting

Figures and tables are well-labeled, but ensure that they are referenced appropriately in the text.

Experimental design

Provide more critical analysis and synthesis of existing literature to highlight the paper's contribution to the field.

Validity of the findings

The methods section is well-detailed, but it would be beneficial to include more information about the rationale behind selecting specific methodologies.

Additional comments

The results are presented clearly, but consider adding more contextual information to help readers understand the significance of the findings.
The paper contributes valuable insights to the field, but refining certain aspects will strengthen its impact.
Ensure consistency in citation style throughout the paper.

Reviewer 3 ·

Basic reporting

The authors should review my comments thoroughly.

Experimental design

The authors should review my comments thoroughly.

Validity of the findings

The authors should review my comments thoroughly.

Additional comments

1. The reasoning behind the identification of the solution lacks robust support and fails to provide convincing literature-based justification. To strengthen the credibility of the proposed hybrid solution, the authors should thoroughly review existing literature and clearly articulate why these specific methods were chosen and integrated. A stronger foundation in the literature would enhance the persuasiveness of the proposed approach.

2. Expanding the experimental scope to include additional algorithms or approaches could potentially lead to better results. By incorporating a broader range of experiments, the authors can provide a more comprehensive understanding of the effectiveness of various approaches and enhance the robustness of the study's findings.

3. Detailed explanations regarding the data preprocessing procedures applied to the dataset are necessary for transparency and reproducibility. It is essential for the authors to clearly outline the steps taken to preprocess the data for their experiments, including any transformations or treatments applied. These steps can be a make or break for the results, and should be clear and reproducible.

4. Updating references to the CSA and Differential algorithms to reflect their current significance in the field is crucial. Integrating more recent literature that highlights the ongoing relevance and advancements of these algorithms would strengthen the theoretical foundation of the study.

5. Conducting a thorough analysis, including an ablation study, would provide valuable insights into the individual contributions of each component of the proposed algorithm to overall performance improvement. Additionally, exploring alternative algorithm pairings instead of LR would offer valuable insights into the versatility and adaptability of the approach.

6. A comparative analysis with simpler algorithms employing more common optimization techniques is necessary to establish the superiority of the proposed CSA-DE-LR approach. By comparing the performance of the proposed solution with existing methods, the authors can demonstrate its competitive advantage.

7. Given that Vanilla LDA and MLP outperform LR in the experiments, it raises questions as to why the authors did not explore incorporating these algorithms into the CSA-DE framework. Exploring alternative hybrid combinations, such as CSA-DE-LDA or CSA-DE-MLP, could potentially lead to further performance improvements.

8. Including an evaluation of the time, space, and computational complexity of the proposed solution compared to existing methods is essential. Providing insights into the resource requirements and scalability of the approach would facilitate a more comprehensive assessment of its practical utility.

9. Incorporating more recent literature to underscore the significance and relevance of the study within the current research landscape is recommended. Updating references to include recent publications would demonstrate the study's alignment with contemporary developments in the field.

10. Conducting an in-depth literature review is crucial to justify the study and strengthen its theoretical underpinnings. By thoroughly reviewing existing literature, the authors can establish the necessity and novelty of their research within the broader context of the field.

11. Demonstrating the generalizability of the solution by applying it to other datasets and assessing its performance would enhance its credibility and applicability in real-world scenarios.

12. Providing clarity on how the complexity of the proposed solution scales with dataset size is essential. Calculating and reporting the current complexity in relation to the presented results would help readers understand the computational demands of implementing the solution in practical settings.

Reviewer 4 ·

Basic reporting

'no comment'

Experimental design

• Provide additional details on the validation process, including statistical analysis, cross-validation techniques used, and comparisons with a broader range of existing methods. This would help to underscore the model’s superiority and reliability.
• Devote a section to exploring methods and techniques for improving the interpretability of the CSA-DE-LR model, such as the use of model explanation tools or simplifying the model architecture without significantly compromising performance.
• Include a discussion on the ethical implications of deploying machine learning models in healthcare, considering aspects like data privacy, potential biases in the model, and the importance of human oversight in decision-making processes.

Validity of the findings

Conduct robust statistical analyses and clearly explain the significance of the results. Provide contextual interpretation to aid understanding.

Increase the sample size or diversify the sample to enhance representativeness. Consider expanding the scope of the study to capture a broader range of perspectives or contexts.

Threats to external validity need to be addressed properly.

generalizability of experimental findings is missing

Additional comments

Conduct a comprehensive review of current and relevant literature to situate your research within the existing body of knowledge, identify gaps that your study aims to fill, and build upon previous work.

Clearly articulate the theoretical or conceptual framework guiding the study, explaining how it informs the research questions, design, analysis, and interpretation of results.

Highlight the unique aspects of your research and its contribution to advancing knowledge in the field. This could be in the form of new findings, methodological innovations, or novel applications of existing theories or technologies.

Discuss how the findings can be applied in real-world settings or contribute to policy, practice, or further research. Provide specific recommendations or guidelines when possible.

---

## Round 0.2 · accepted · Accept

I confirm that the Authors have addressed the reviewer comments property.

Reviewer 3 ·

Basic reporting

The author seems to have addressed most of my concerns adequately. I can recommend it for publication but necessary proofreading and critical validation must be done once more.

Experimental design

The author seems to have addressed most of my concerns adequately. I can recommend it for publication but necessary proofreading and critical validation must be done once more.

Validity of the findings

The author seems to have addressed most of my concerns adequately. I can recommend it for publication but necessary proofreading and critical validation must be done once more.

Additional comments

The author seems to have addressed most of my concerns adequately. I can recommend it for publication but necessary proofreading and critical validation must be done once more.